# G protein subunit Gγ13-mediated signaling pathway is critical to the inflammation resolution and functional recovery of severely injured lungs

Yi-Hong Li[1], Yi-Sen Yang[1], Yan-Bo Xue[1], Hao Lei[1], Sai-Sai Zhang[1], Junbin Qian[2,3,4], Yushi Yao[5], Ruhong Zhou[1,6]*, Liquan Huang[1,6,7]*

[1]College of Life Sciences, Zhejiang University, Hangzhou, China; [2]Zhejiang Provincial Key Laboratory of Precision Diagnosis and Therapy for Major Gynecological Diseases, Women's Hospital, Zhejiang University School of Medicine, Hangzhou, China; [3]Institute of Genetics, Zhejiang University School of Medicine, Hangzhou, China; [4]Cancer Center, Zhejiang University, Hangzhou, China; [5]Institute of Immunology and Sir Run Run Shaw Hospital, Zhejiang University School of Medicine, Hangzhou, China; [6]Zhejiang University Shanghai Institute for Advanced Study, Shanghai, Shanghai, China; [7]Monell Chemical Senses Center, Philadelphia, United States

*For correspondence:
rhzhou@zju.edu.cn (RZ);
huangliquan@zju.edu.cn (LH)

Competing interest: The authors declare that no competing interests exist.

**Abstract** Tuft cells are a group of rare epithelial cells that can detect pathogenic microbes and parasites. Many of these cells express signaling proteins initially found in taste buds. It is, however, not well understood how these taste signaling proteins contribute to the response to the invading pathogens or to the recovery of injured tissues. In this study, we conditionally nullified the signaling G protein subunit Gγ13 and found that the number of ectopic tuft cells in the injured lung was reduced following the infection of the influenza virus H1N1. Furthermore, the infected mutant mice exhibited significantly larger areas of lung injury, increased macrophage infiltration, severer pulmonary epithelial leakage, augmented pyroptosis and cell death, greater bodyweight loss, slower recovery, worsened fibrosis and increased fatality. Our data demonstrate that the Gγ13-mediated signal transduction pathway is critical to tuft cells-mediated inflammation resolution and functional repair of the damaged lungs. To our best knowledge, it is the first report indicating subtype-specific contributions of tuft cells to the resolution and recovery.

## eLife assessment

This, in principle, **useful** study suggests that the G-protein subunit Gng13 is required for limiting injury and inflammation following H1N1 influenza infection via anti-inflammatory effects from ectopic tuft cells. While support for Gng13 helping to limit influenza injury in the transgenic mouse models used here is **solid**, evidence for these effects being mediated by normal tuft cells remains **incomplete**, giving conflicting data from mice that lack tuft cells entirely.

## Introduction

The respiratory system comprises the nasal cavity, airways, and lungs, and has the largest surface area exposed to the external environment. It is thus constantly challenged by various stimuli, ranging from odorants, pollutants, toxins, allergens, to viruses, bacteria, and fungi. Prompt detection and appropriate responses to harmful inhalants are critical to maintaining not only the pivotal gas-exchange

function of the system but also the overall health or even vitality of an individual. How the host responds to the external challenges and what damages can be inflicted upon the host depend on the types of harmful stimulus, dosage, host genetic background, and immune history (*Yunis et al., 2023*).

Over the past decades, viral infection has been salient risks to human health. The severe acute respiratory syndrome coronavirus (SARS-CoV-1), and H1N1 influenza A virus and SARS-CoV2 have caused pandemics (*Peiris, 2003*; *Donaldson et al., 2009*; *Chen et al., 2020*). However, it remains elusive how the host and infectious viruses interact and how the host regulates its responses to the invasion. Both under- and over-reactions by the host appear to engender serious consequences.

Recent studies have shown that several types of epithelial cells expressing the chemosensory receptors play important roles in detecting and responding to the external stimuli in the respiratory system (*Carey and Lee, 2019*; *Tizzano and Finger, 2013*). Among them is a rare type of microvillous cells, tuft cells (*Billipp et al., 2021*), which have also been found in a number of tissues, including the gastrointestinal tract, pancreas, respiratory airways and lungs, gallbladder, and thymus (*Bezençon et al., 2008*; *Miller et al., 2018*; *Montoro et al., 2018*; *Nevalainen, 1977*; *Saqui-Salces et al., 2011*; *DelGiorno et al., 2020*). These cells can sense noxious stimuli and invading pathogens, such as allergens, bacteria, protists, and helminths (*Gerbe et al., 2016*; *Howitt et al., 2016*; *Lei et al., 2018*; *Luo et al., 2019*; *von Moltke et al., 2016*). Tuft cells are known to express canonical chemosensory signaling pathways, including the sweet and umami taste receptor subunit Tas1r3 (*Howitt et al., 2020*), bitter taste receptors Tas2rs (*Luo et al., 2019*; *Imai et al., 2020*), and olfactory receptors Vmn2r26 (*Xiong et al., 2022*), as well as their downstream signaling proteins including the heterotrimeric G protein subunits $\alpha$-gustducin, G$\beta$1 and G$\gamma$13, phospholipases, and the transient receptor potentialion channel Trpm5 (*Howitt et al., 2016*; *Doyle et al., 2023*). Even the signal transduction of the Sucnr1 receptor for succinic acid and free fatty acid receptor 2 Ffar2 for propionate, characteristic metabolites of certain microbes, is mediated by $\alpha$-gustducin and/or Trpm5 signaling proteins to monitor changes in gut microbiota and airway infection. And ablation of either $\alpha$-gustducin or Trpm5 diminishes the intestinal and tracheal epithelia's responses to the infection of the parasitic helminths and microbial colonization (*Lei et al., 2018*; *Nadjsombati et al., 2018*; *Schneider et al., 2018*; *Keshavarz et al., 2022*; *Perniss et al., 2023*).

Once stimulated, tuft cells can release a number of output signaling molecules, including IL-25, prostaglandins (*Kotas et al., 2022*), cysteinyl leukotrienes (*Bankova et al., 2018b*; *Bankova and Boyce, 2018a*; *McGinty et al., 2020*), acetylcholine (*Hollenhorst et al., 2020*; *Krasteva et al., 2011*), and other substances, to stimulate type 2 innate lymphoid cells, nerve terminals or adjacent cells, initiating innate and adaptive immune responses, leading to the elimination of irritants and pathogens, and eventually to tissue repair and remodeling (*Finger et al., 2003*; *Tizzano et al., 2010*; *Saunders et al., 2014*). However, tuft cells are heterogeneous in terms of gene expression patterns, signal transduction pathways, and output effectors (*Banerjee et al., 2018*). For example, some tuft cells express subsets of Tas2r receptors, or cysteinyl leukotriene receptor 3 (Cysltr3) while others do not (*Luo et al., 2019*; *Imai et al., 2020*; *Bankova et al., 2018b*). It seems important to determine ligand profiles, output signals, and physiological roles for tuft cells residing in different tissues to fully understand their protective functions against various pathogens and at different stages of disease progression.

Although they are normally only found in the nasal cavity and trachea, rarely found distal to the lung hilus (*Krasteva et al., 2011*; *Tizzano et al., 2010*; *Saunders et al., 2014*), tuft cells can be ectopically formed in the distal lung following severe viral infection or chemical damage (*Rane et al., 2019*; *Barr et al., 2022*; *Roach et al., 2022*; *Melms et al., 2021*). These ectopic tuft cells of molecular diversities emerge around 12 days post-viral infection, suggesting that they may not play any role in the initial response to the infection or during viral clearance (*Rane et al., 2019*; *Barr et al., 2022*). However, it is unclear whether each subtype of these diverse tuft cells makes unique contributions to the subsequent inflammation resolution, tissue repair, and remodeling. In this study, we conditionally nullified the expression of a heterotrimeric G protein subunit, G$\gamma$13, in the choline acetyltransferase (ChAT)-expressing tuft cells, which resulted in the generation of fewer tuft cells, but conferred even severer injury, slower recovery, and higher fatality following the H1N1 influenza infection. Our data indicate that subsets of ectopic tuft cells with or without G$\gamma$13 expression play different important roles in the inflammation resolution and tissue repair.

## Results

### Severe injury induces the generation of taste signaling proteins-expressing ectopic tuft cells

To verify the previously reported ectopic generation of lung tuft cells upon severe injury, we intranasally inoculated adult mice with a sublethal dose of H1N1 viruses, and observed the bodyweight changes over a course of 25 days post infection (dpi) (*Figure 1A*). The mice were then killed and their lungs were dissected out for analyses (*Figure 1A*). The mice showed bodyweight loss over the days post-infection with a peak loss of up to 25% of their original bodyweight around 8–9 dpi, and their lungs displayed lesion areas, of which the histological analysis showed massive alveolar damage and severe epithelial dysplasia with altered tissue structures as well as an increased number of CD45$^+$ immune cells in comparison with the uninfected control (*Figure 1B, C and D*). Analysis of the H1N1 injured-lungs of the *Chat*-Cre: Ai9 mice, which express tdTomato proteins in the ChAT-expressing cells, showed that about 78% of ChAT-expressing cells also expressed the tuft cell marker Dclk1 whereas about 60% and 12% of ChAT-expressing cells expressed the other two taste signaling proteins G$\alpha$-gustducin and Plc$\beta$2, respectively (*Figure 1E, F*). In addition, chemically induced severe lung injury with bleomycin, lipopolysaccharide, and powder of house dust mite extracts also led to the generation of dysplastic tuft cells, but to a lesser extent (*Figure 1—figure supplement 1*), confirming the severe injury-induced formation of ectopic tuft cells in the lung parenchyma (*Rane et al., 2019*; *Barr et al., 2022*; *Huang, 2022*).

### Ectopic lung tuft cells respond to bitter tasting compounds

To assess how the severe lung injury may affect the bitter taste receptor expression, we quantified and comparatively analyzed Tas2r expression levels by performing quantitative reverse transcription-PCR on the WT lung tissues before and after H1N1 infection. The results showed that the expression levels of most of murine 35 Tas2rs remained unchanged while *Tas2r105, Tas2r108, Tas2r118, Tas2r137,* and *Tas2r138* were upregulated, and *Tas2r135* and *Tas2r143* were downregulated (*Figure 2A*), which is largely consistent with the single-cell RNAseq data (*Figure 2—figure supplement 1*; *Barr et al., 2022*), suggesting that the upregulated Tas2rs are indeed expressed in these ectopic tuft cells.

To determine whether the taste signal transduction pathways present in these ectopic lung tuft cells are functional, we isolated these tuft cells from the *Chat*-Cre: Ai9 mice at 25 dpi by fluorescence-activated cell sorting (FACs) (*Figure 2—figure supplement 2*) and assessed their intracellular calcium responses to two common bitter taste substances denatonium benzoate (D.B.) and quinine. It is known that D.B. activates mouse bitter taste receptor Tas2r105 whereas quinine can stimulate multiple Tas2rs, including 3 of the 5 upregulated bitter taste receptors: Tas2r105, Tas2r108, and Tas2r137 (*Lossow et al., 2016*). The results showed that subsets of tuft cells responded to these two compounds by increasing the intracellular calcium concentrations. Application of the bitter taste inhibitor allyl isothiocyanate (AITC), G protein $\beta\gamma$ moiety inhibitor gallein or Plc$\beta$2 inhibitor U73122 was able to completely inhibit these cells' calcium responses to the bitter compounds (*Figure 2B-K*), indicating that the Tas2rs and their downstream signaling proteins are functional in subsets of tuft cells, and may play some physiological roles during the transition from inflammatory response to resolution.

### Conditional nullification of the taste signaling protein G$\gamma$13 exacerbates the H1N1-inflicted disease severity and suppresses ectopic tuft cell expansion

To determine what role the taste signaling proteins may play in the H1N1 infection-triggered lung injury, we inoculated H1N1 to the whole-body *Trpm5* knockout mice (*Trpm5$^{-/-}$*) and the conditional G$\gamma$13 knockout mice (*Chat*-Cre: *Gng13$^{flox/flox}$*, i.e., *Gng13*-cKO) as well as WT control mice. Results indicated that while WT and *Trpm5$^{-/-}$* mice showed similar bodyweight loss/recovery curves with nearly no mortality, *Gng13*-cKO mice exhibited significantly more bodyweight loss with a maximum of 30% instead of 20% of the original bodyweight, and a slower recovery process with a shifted maximum bodyweight loss from 9 dpi to 10 dpi, and a significant increase in the mortality rate to 37.5%, the majority of which occurred between 8 and 12 dpi (*Figure 3A, B*). Furthermore, the H1N1-infected lungs of WT and *Trpm5$^{-/-}$* mice at 25 dpi displayed similar injured surface areas and damaged tissue volumes whereas those of *Gng13*-cKO showed significantly greater injured areas and volumes

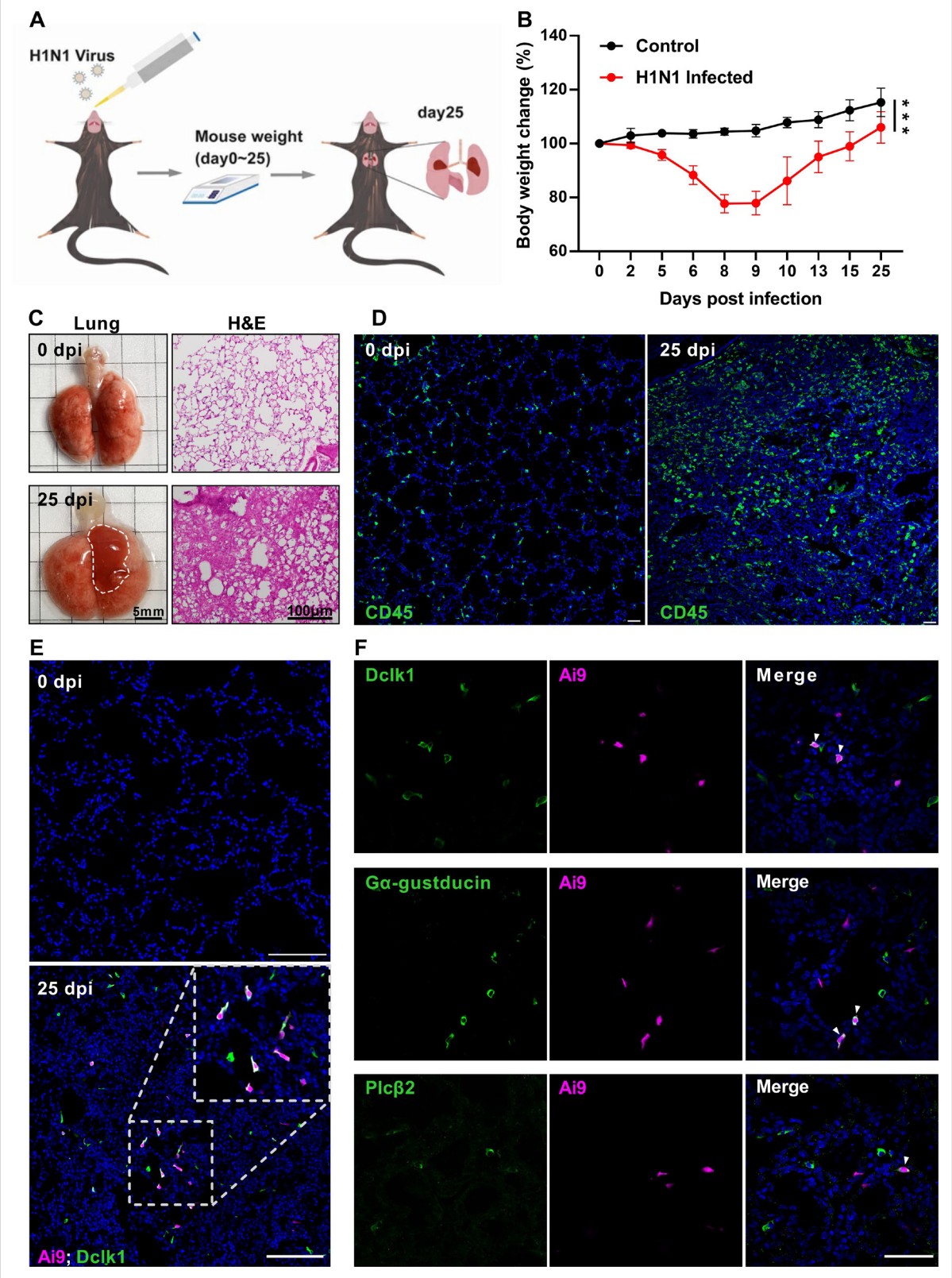

**Figure 1.** H1N1 infection causes severe damage to mouse lung. (**A**) Schematic of H1N1 intranasal inoculation procedure. (**B**) Bodyweight changes of uninfected control and H1N1-infected mice over 25 days post infection. Data are presented as means ± SD (n=6). Unpaired two-tailed student t-tests were performed. (**C**) Representative images of lungs at 0 and 25 dpi, and their hematoxylin and eosin (H&E) tissue sections (Bars: 5 mm in the whole lung images and 100 μm in the lung section images). (**D**) Immunostaining images of lung sections of 0 and 25 dpi with an antibody to CD45.

*Figure 1 continued on next page*

*Figure 1 continued*

(**E**) Immunostaining of the *Chat*-Cre: Ai9 lung sections of 0 and 25 dpi with the antibody to the tuft cell marker Dclk1 (green), and Ai9 (red tdTomato). (**F**) Immunostaining of the *Chat*-Cre: Ai9 lung sections of 25 dpi with antibodies to Dclk1 (green), Gα-gustducin (green) or Plcβ2 (green) and Ai9 (red tdTomato). Scale bars in (**D, E, F**): 50 μm. ***p<0.001.

The online version of this article includes the following source data and figure supplement(s) for figure 1:

**Source data 1.** Mouse bodyweight loss following H1N1 infection.

**Figure supplement 1.** Chemically induced severe injury also engenders ectopic formation of lung tuft cells.

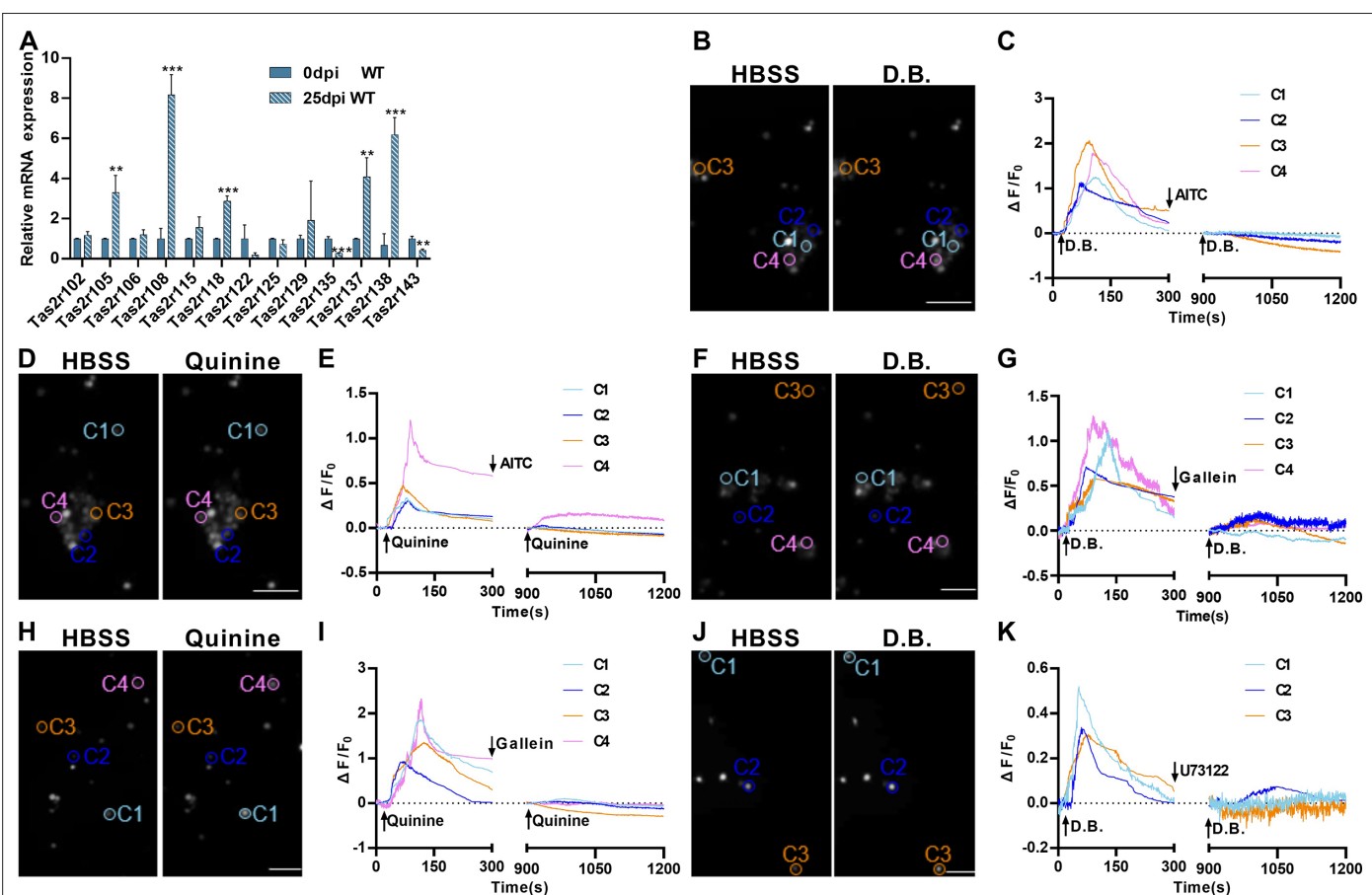

**Figure 2.** Expression and functional responses of bitter taste receptors. (**A**) Quantitative RT-PCR analysis of bitter taste receptor expression in the lung tissues at 0 dpi versus 25 dpi. Expression of *Tas2r105*, *Tas2r108*, *Tas2r118*, *Tas2r137*, and *Tas2r138* was upregulated whereas *Tas2r135* and *Tas2r143* were downregulated at 25 dpi. Data are presented as means ± SD (n=3). Unpaired two-tailed student t-tests were performed. (**B–K**) Calcium imaging and response traces of FACs-sorted tuft cells isolated from H1N1-infected lungs at 25 dpi. (B, D, F, H, and J) Left panels are gray images of tuft cells at rest in the HBSS buffer whereas the right panels are gray images of the same cells after being stimulated by denatonium benzoate (D.B.) or quinine. Activated cells appeared brighter. (**C, E, G, I, K**) traces of calcium responses of tuft cells on the left. Some tuft cells responded to D.B, which was inhibited by allyl isothiocyanate (AITC) (**B, C**), gallein (**F, G**), or U73122 (**J, K**) whereas others responded to quinine, which was also inhibited by AITC (**D, E**), or gallein (**H, I**). Scale bars: 50 μm. **p<0.01, ***p<0.001.

The online version of this article includes the following source data and figure supplement(s) for figure 2:

**Source data 1.** Tas2r expression in the lungs and intracellular calcium responses of tuft cells to bitter substances.

**Figure supplement 1.** Expression of bitter taste receptors in tuft cell clusters.

**Figure supplement 2.** Fluorescence-activated cell sorting (FACs) sorting of ectopic tuft cells.

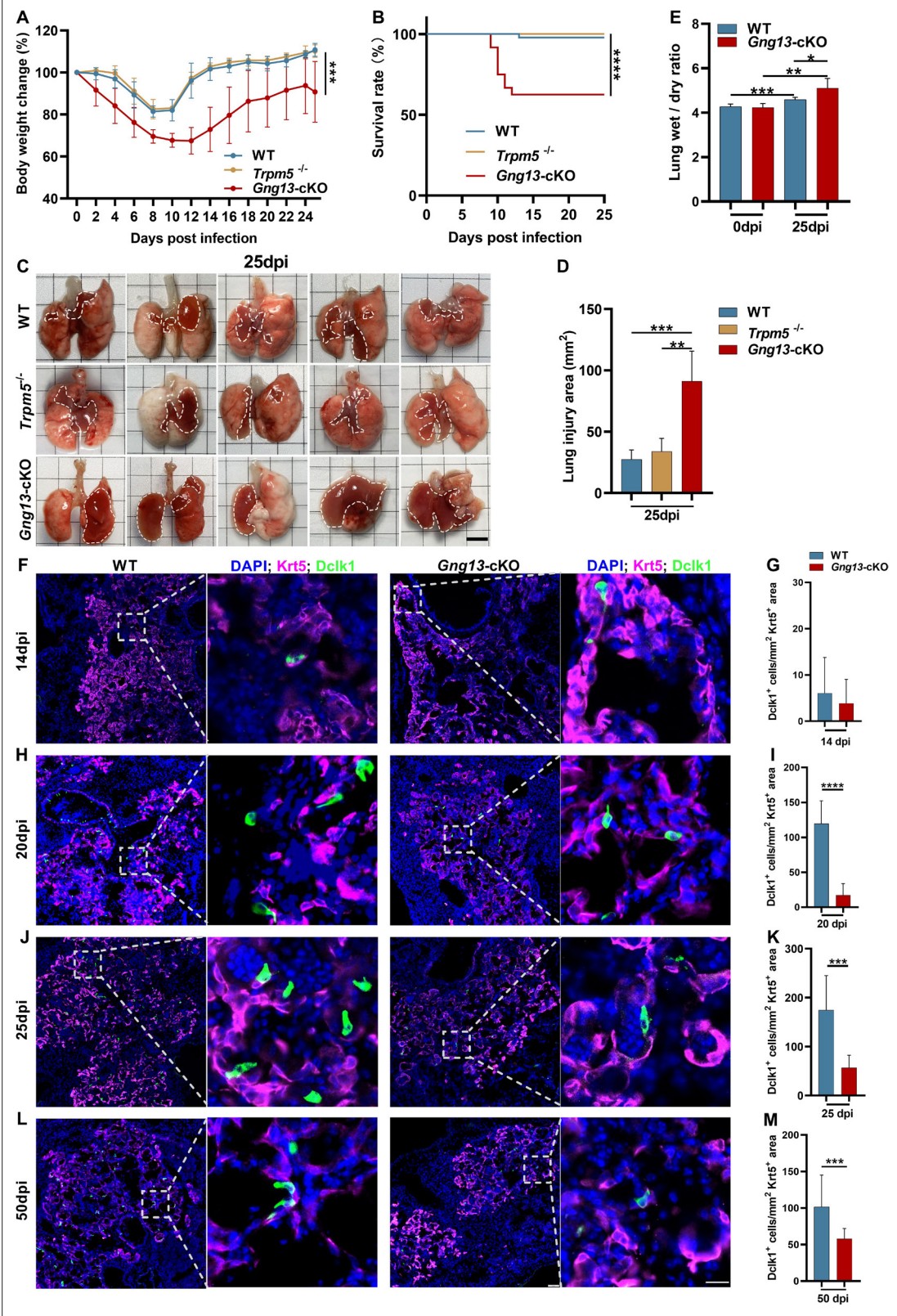

**Figure 3.** H1N1 infection inflicts differential lung injury and tuft cell dysplasia on wild-type (WT) versus mutant mice. (**A**) *Gng13*-cKO mice showed greater bodyweight loss than WT or *Trpm5*-/- mice over days post-infection. Data are presented as means ± SD (n=6). Unpaired two-tailed t-tests were performed. (**B**) Kaplan-Meier survival curves of WT, *Trpm5*-/- and *Gng13*-cKO mice following H1N1 inoculation. A significant number of *Gng13*-cKO mice died between 9 and 12 dpi (i.e., 2, 4, 2 mice, and 1 mouse died at 9, 10, 11, 12 dpi, respectively), reducing the overall survival rate to 62.5%, significantly

*Figure 3 continued on next page*

*Figure 3 continued*

lower than those of *Trpm5*[-/-] or WT mice, of which nearly all survived. Data are presented as means ± SEM (n=44, 17 and 14 for WT, *Trpm5*[-/-], and *Gng13*-cKO mice, respectively). The curves were determined by a log-rank test. (**C, D**) Images and statistical analysis of the injured areas on the lungs of WT, *Trpm5*[-/-], and *Gng13*-cKO mice at 25 dpi. The injured areas that are marked by dashed lines were significantly greater in the *Gng13*-cKO mice than in *Trpm5*[-/-] or WT mice while no significant difference was found between the latter two. Data are presented as means ± SD (n=5), and unpaired two-tailed student t-tests were performed. Scale bar: 5 mm. (**E**) Comparative analysis of wet-to-dry weight ratios of WT and *Gng13*-cKO lungs at 0 and 25 dpi. While no difference in the ratio was found between WT and *Gng13*-cKO mice at 0 dpi, the ratios of these mice at 25 dpi were significantly greater than their corresponding ones at 0 dpi. However, the *Gng13*-cKO ratio was even greater than the WT ratio at 25 dpi. Data are presented as means ± SD (n=6), and unpaired two-tailed student t-tests were performed. (**F–M**) Identification of tuft cells in the Krt5-expressing tissue areas of H1N1-injured lungs of WT and *Gng13*-cKO mice at 14, 20, 25, and 50 dpi using antibodies to Krt5 and to the tuft cell marker Dclk1. The densities of tuft cells were significantly higher in WT than in *Gng13*-cKO lungs at 20, 25, and 50 dpi, but not at an earlier time point of 14 dpi. Data are presented as means ± SD (n=3), and unpaired two-tailed student t-tests were performed. Scale bars in (L): 50 μm and 15 μm. *p<0.05, **p<0.01, ***p<0.001, ****p<0.0001.

The online version of this article includes the following source data and figure supplement(s) for figure 3:

**Source data 1.** Bodyweight changes, survival rates, lung injury areas and volumes, lung wet/dry ratios, and the number of tuft cells in the injured areas of WT, *Trpm5*[-/-], and *Gng13*-cKO mice.

**Figure supplement 1.** Statistical analysis of the injured tissue volumes on the lungs of wild-type (WT), *Trpm5*[-/-], and *Gng13*-cKO mice at 25 dpi.

**Figure supplement 1—source data 1.** Lung injury volumes of wild-type (WT), *Trpm5*[-/-], and *Gng13*-cKO mice.

**Figure supplement 2.** Ectopic generation of tuft cells in wild-type (WT), *Trpm5*[-/-], and *Gng13*-cKO mice over days post H1N1 infection.

**Figure supplement 2—source data 1.** The numbers of tuft cells in the injured areas of WT, Trpm5 and Gng13-cKO mice, and gene expression analyses of a single tuft cell RNAseq dataset.

(***Figure 3C, D, Figure 3—figure supplement 1***). And the ratios of wet-to-dry lung masses of both WT and *Gng13*-cKO mice at 25 dpi were significantly greater than their respective ones at 0 dpi whereas the ratio of the *Gng13*-cKO mice at 25 dpi was even greater than that of the WT mice at 25 dpi (***Figure 3E***), supporting the notion that *Gng13*-cKO mice experienced severer damage following H1N1 infection compared with WT control.

To investigate how the abolishment of taste signaling proteins affects the formation of ectopic lung tuft cells in response to H1N1 infection, we performed immunohistochemistry on lung tissue sections, using an anti-Krt5 antibody to demarcate H1N1-damaged areas and an anti-Dclk1 antibody to identify tuft cells. We found as early as 14 dpi a small number, i.e., ~6 and ~4 tuft cells per mm$^2$ of the Krt5-expressing tissue area in WT and *Gng13*-cKO lungs, respectively, and no significant differences in the tuft cell density between these two genotypes were found at this time point. However, at 20 and 25 dpi, the densities of tuft cells in WT were increased to 120 and 175 cells per mm$^2$, respectively, which are significantly more than the corresponding densities of *Gng13*-cKO, i.e., 17 and 57 cells/mm$^2$, respectively; and at 50 dpi, the tuft cell density in WT was decreased to 102 cells/mm$^2$, which, however, was still significantly higher than 56 cells/mm$^2$ of *Gng13*-cKO (***Figure 3F-M***). In contrast, the densities of tuft cells in *Trpm5*[-/-] mice were nearly identical to those of WT at both 14 and 25 dpi, and no significant difference was found between WT and *Trpm5*[-/-] mice (***Figure 3—figure supplement 2A-E***). Thus, although tuft cells first appeared at 14 dpi in all WT, *Trpm5*[-/-] and *Gng13*-cKO mice, and no statistic difference in their densities was found among these mice, the subsequent densities were increased in all WT, *Trpm5*[-/-], and *Gng13*-cKO mice; and the increase was much less in the *Gng13*-cKO mice than in WT or *Trpm5*[-/-], and consequently, the densities in the *Gng13*-cKO mice were significantly less than in WT or *Trpm5*[-/-] mice at 20, 25, and 50 dpi (***Figure 3—figure supplement 2F***).

To determine how many ectopic tuft cells in the infected WT lung express Gγ13 and whether any remaining ectopic tuft cells in the infected *Gng13*-cKO lung express Gγ13, double immunostaining was carried out with antibodies to Dclk1 and to Gγ13 on the lung tissue sections of 25 dpi of both WT and *Gng13*-cKO mice. The results showed that about 28.6% of Dclk1[+] ectopic tuft cells express Gγ13 in WT sections while none of the remaining Dclk1[+] tuft cells in the *Gng13*-cKO sections was Gγ13[+] (***Figure 3—figure supplement 2G, H***), indicating an effective conditional ablation of Gγ13 expression. Furthermore, reanalysis of the previously published single tuft cell RNAseq dataset GSE197163 showed that about 57% of the Trpm5-GFP[+] tuft cells were Gγ13[+] in WT mice, some of which also expressed Alox5, a key enzyme to the biosynthesis of pro-resolving mediators (***Figure 3—figure supplement 2I,J***; ***Barr et al., 2022***; ***Halade et al., 2022***). This result is consistent with the previous report indicating the heterogeneity of dysplastic tuft cells in the severely injured lungs, and this study ablated the Gγ13[+] tuft cells including some expressing the pro-resolving enzyme Alox5.

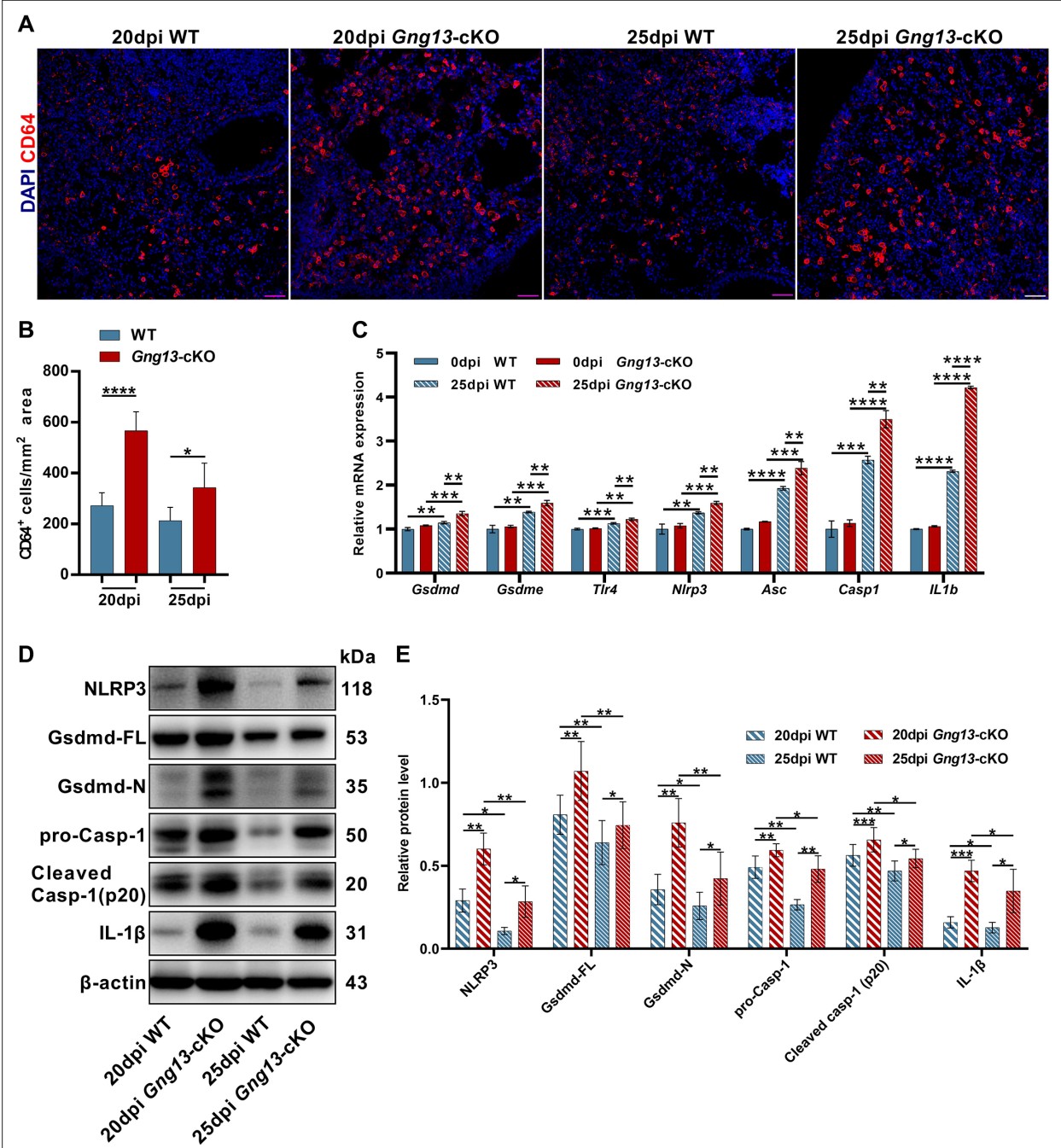

**Figure 4.** Stronger inflammatory response and upregulated expression of pyroptotic genes in the H1N1-infected *Gng13*-cKO lungs. (**A, B**) Immunostaining of wild-type (WT) and *Gng13*-cKO lung tissue sections with an antibody to the immune cell marker CD64 indicates significantly more immune cells in the mutant lung than in WT at both 20 and 25 dpi. Scale bar: 50 μm. Data are presented as means ± SD (n=3), and unpaired two-tailed student t tests were performed. (**C**) qRT-PCR indicates that expression levels of gasdermin D and E (*Gsdmd, Gsdme*) and other proteins of the pyroptosis pathway: *Tlr4, Nlpr3, Asc, Casp1,* and *Il1b* were significantly upregulated in both WT and *Gng13*-cKO lungs at 25 dpi compared with those at 0 dpi. Furthermore, the expression levels of these genes in *Gng13*-cKO were significantly higher than in WT at 25 dpi. Data are presented as means ± SD (n=3). Unpaired two-tailed student t-tests were performed. (**D, E**) Western blot analysis shows that the protein levels of NLRP3, full-length Gasdermin D (Gsdmd-FL), N-terminal fragment of Gasdermin D (Gsdmd-N), pro-caspase 1 (pro-Casp1), cleaved caspase 1 (Cleaved Casp-1 (p20)), and IL-1$\beta$ in the *Gng13*-cKO lungs were significantly higher than the corresponding levels of WT at both 20 and 25 dpi ($\beta$-actin as an internal control). And over this phase, the expression levels of these proteins were reduced at 25 dpi in comparison with those at 20 dpi in both genotypes. Data are presented as means ± SD (n=4), and unpaired two-tailed t-tests were performed. *p<0.05, **p<0.01, ***p<0.001, ****p<0.0001.

The online version of this article includes the following source data and figure supplement(s) for figure 4:

*Figure 4 continued on next page*

*Figure 4 continued*

**Source data 1.** CD64$^+$ cell densities and pyroptosis-related gene expression in the control and injured lungs of wild-type(WT) and *Gng13*-cKO mice.

**Source data 2.** Original file for the protein expression levels of pyroptotic genes in wild-type (WT) and *Gng13*-cKO mice.

**Source data 3.** Labeled file for the protein expression levels of pyroptotic genes in wild-type (WT) and *Gng13*-cKO mice.

**Figure supplement 1.** Quantitative reverse transcription-PCR (qRT-PCR) analysis of gasdermin gene expression in wild-type (WT) and *Gng13*-cKO lungs.

**Figure supplement 1—source data 1.** Pyroptosis-related gene expression in the control and injured lungs of wild-type (WT) and Gng13-cKO mice.

## Gγ13 mutant mice display stronger inflammation and augmented pyroptosis following H1N1 infection

To determine whether Gγ13 conditional knockout affects the inflammation resolution following H1N1 infection, we performed immunohistochemistry on the lung sections with an antibody to the immune cell marker CD64. The results showed that at both 20 and 25 dpi, the *Gng13*-cKO mice had significantly more CD64$^+$ immune cells in the injured areas than WT, although both types of mice indicated a trend of inflammation resolution with a reduction in the number of immune cells from 20 to 25 dpi (*Figure 4A, B*).

We also carried out quantitative reverse transcription-PCR (qRT-PCR) to assess the expression of pyroptosis-related gasdermin genes, and found that *Gsdmd* and *Gsdme* were the two most abundantly expressed ones (*Figure 4—figure supplement 1*). Further analysis showed that before H1N1 infection, the expression levels of *Gsdmd* and *Gsdme* were similar between WT and *Gng13*-cKO mice at 0 dpi; and at 25 dpi, their expression levels were significantly higher than those at 0 dpi in both WT and *Gng13*-cKO mice; furthermore, *Gsdmd* and *Gsdme* expression levels were significantly higher in the *Gng13*-cKO mice than in WT at 25 dpi (*Figure 4C*). We then performed additional qRT-PCR to examine the expression patterns of other genes involved in the pyroptosis, *Tlr4*, *Nlpr3*, *Asc*, *Casp1* and *Il1b*, and found a similar pattern for all these genes, i.e., both WT and *Gng13*-cKO mice showed more expression at 25 dpi than at 0 dpi, whereas *Gng13*-cKO exhibited even higher expression levels than WT at 25 dpi (*Figure 4C*).

To validate their expression at the protein level, we isolated proteins from the injured lung areas and performed Western blot analysis with antibodies to NLRP3, full-length Gsdmd, N-terminal fragment of Gsdmd, pro-Casp-1, cleaved Casp-1, IL-1β, and β-actin as an internal reference (*Figure 4D*). Quantitative analysis showed that at both 20 and 25 dpi, the protein levels of all these pyroptosis pathway proteins were greater in the *Gng13*-cKO mice than in WT mice, and comparison within the same genotype showed that the levels of these proteins at 20 dpi were higher than at 25 dpi in both WT and *Gng13*-cKO mice (*Figure 4E*).

To determine which cells underwent pyroptosis, we performed immunohistochemistry on the injured lung sections with antibodies to the two most highly expressed gasdermins, Gsdmd, and Gsdme. Results showed that at 25 dpi, there were about 180 and 77 cells/mm$^2$ in *Gng13*-cKO mice expressing Gsdmd and Gsdme, respectively, which are significantly higher than the corresponding densities of 88 and 50 cells/mm$^2$ in WT (*Figure 5A, B*; *Figure 5—figure supplement 1A, B*). Colocalization of immunostaining with antibodies to Gsdmd, the macrophage marker F4/80 and the epithelial cell marker EpCAM, indicated that the majority of Gsdmd$^+$ cells, i.e., about 76.7% and 80.6% in WT and *Gng13*-cKO mice, respectively, at 25 dpi were also F4/80$^+$, and the remaining Gsdmd$^+$ cells, i.e., 21.9% and 17.8% of Gsdmd$^+$ cells in WT and *Gng13*-cKO mice, respectively, were EpCAM$^+$, suggesting that most of Gsdmd-expressing cells were macrophages whereas the remaining were epithelial cells (*Figure 5C*, *Figure 5—figure supplement 2A*). Interestingly, nearly all (i.e. 98.5% and 95.5% in WT and *Gng13*-cKO, respectively) of Gsdme$^+$ cells were F4/80$^+$, indicating that almost all Gsdme-expressing cells were macrophages in the two genotypes at 25 dpi (*Figure 5—figure supplement 2B*). Finally, the density of cells expressing IL-1β and caspase 3 was 153 cells/mm$^2$ and 267 cells/mm$^2$, respectively, in *Gng13*-cKO mice at 25 dpi, and significantly greater than the corresponding density of 63 and 66 cells/mm$^2$ in WT mice (*Figure 5D, E*; *Figure 5—figure supplement 1C, D*), indicating more programmed cell death in the H1N1-infected *Gng13*-cKO lung. Similar to that of Gsdme, the majority, i.e., 93% and 91%, of IL-1β-expressing cells in WT and *Gng13*-cKO mice, respectively, were F4/80$^+$ macrophages (*Figure 5—figure supplement 2C*).

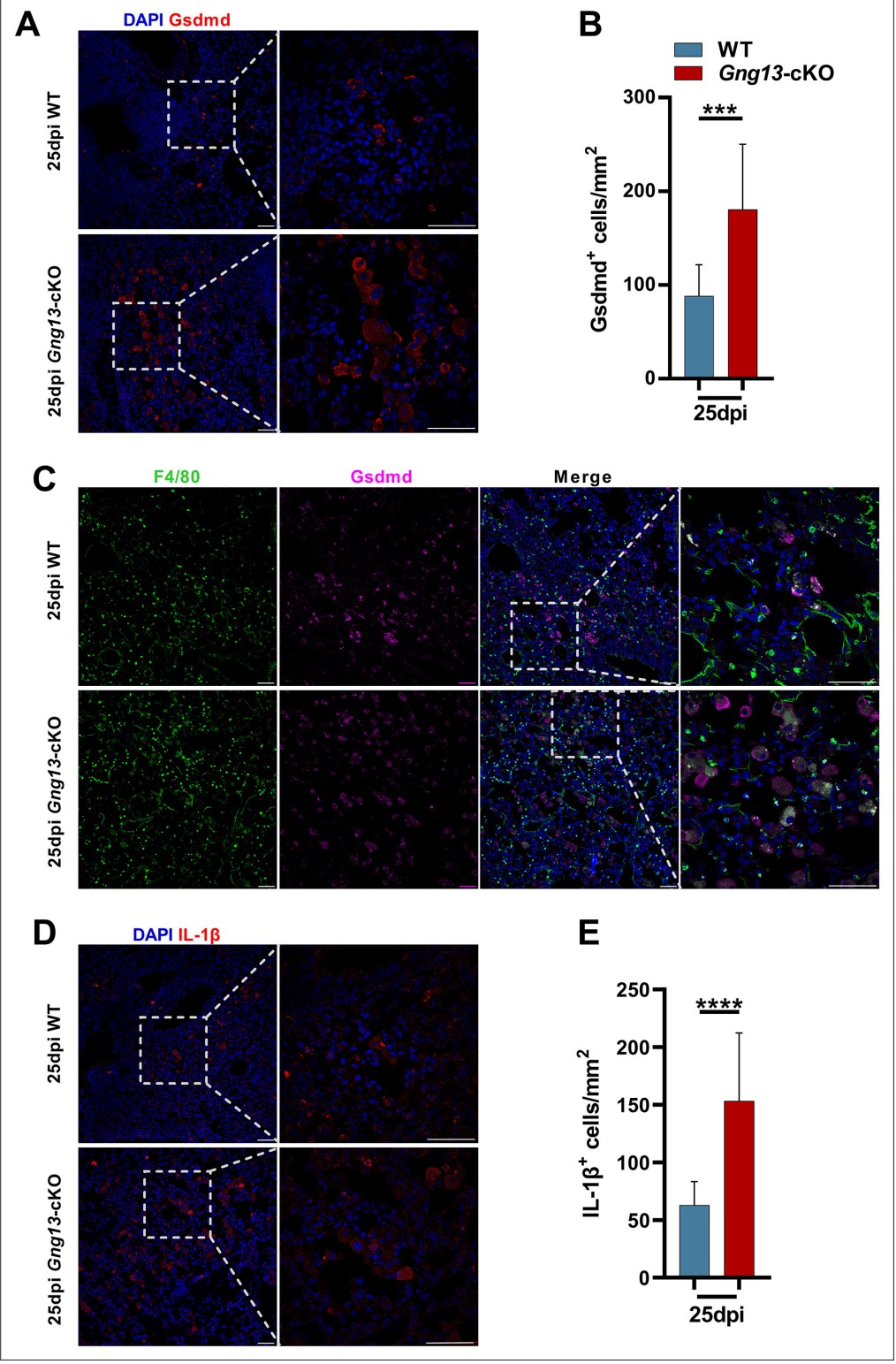

**Figure 5.** Immunohistochemical analyses of gasdermin D and IL-1β expressing cells. (**A, B**) Immunostaining of H1N1-infected lung sections antibody to gasdermin D (Gsdmd) indicates significantly more gasdermin D-expressing cells in the *Gng13*-cKO sections than in wild-type (WT). Data are presented as means ± SD (n=3), and unpaired two-tailed student t-tests were performed. (**C**) Co-immunostaining shows that nearly the same percentage, i.e., 98.5% and 95.5% of Gsdmd+ cells in WT and *Gng13*-cKO, respectively, were F4/80+. (**D**) Immunostaining of H1N1-infected lung sections at 25 dpi with an antibody to IL-1β indicate significantly

*Figure 5 continued on next page*

*Figure 5 continued*

more IL-1$\beta$-expressing cells in the *Gng13*-cKO sections than in WT. Data are presented as means ± SD (n=4), and unpaired two-tailed t-tests were performed. Scale bars: 50 µm. ***p<0.001, ****p<0.0001.

The online version of this article includes the following source data and figure supplement(s) for figure 5:

**Source data 1.** The densities of Gsdmd$^+$, IL-1$\beta^+$ cells in the injured lung tissues of wild-type (WT) and *Gng13*-cKO mice.

**Figure supplement 1.** Expression of gasdermin E and caspase 3 in the H1N1-injured lungs.

**Figure supplement 1—source data 1.** The densities of gasdermin E (Gsdme) and Caspase-3$^+$ cells in the injured lung tissues of wild-type (WT) and Gng13-cKO mice.

**Figure supplement 2.** Partial colocalization of gasdermin D, E, and IL-1$\beta$ to epithelial cells and macrophages.

To obtain a snapshot of cells indeed undergoing pyroptosis, we intranasally administrated Sytox to label dead cells (*Xi et al., 2021*). Histochemical analysis showed that *Gng13*-cKO mice at 20 and 25 dpi had 211 and 96 Sytox$^+$ cells/mm$^2$, respectively, significantly more than the corresponding 96 and 62 cells/mm$^2$ in WT (*Figure 6A, B*). Double staining of Sytox with an anti-CD64 antibody indicated that more than half of CD64$^+$ macrophages, i.e., 61.5% and 57% in WT and *Gng13*-cKO mice at 25 dpi, respectively, were Sytox$^+$, undergoing pyroptosis (*Figure 6C*). Together, these data indicate that the injured lungs exhibited immune cell infiltration, expressed pyroptotic genes, and cell death of macrophages and some epithelial cells; and from 20 to 25 dpi, the injured animals showed some inflammation resolution; but at both time points, the *Gng13*-cKO mice showed stronger inflammatory responses and slower inflammation resolution than WT.

## G$\gamma$13 disruption leads to severer leakage of the lung epithelia

Increased pyroptosis in the mutant lungs prompted us to assess the integrity of lung epithelia. We collected total bronchoalveolar lavage fluid (BALF) from both WT and *Gng13*-cKO mice at 0, 20, and 25 dpi, and the average numbers of cells per mouse present in the BALF from WT and *Gng13*-cKO mice at 0 dpi were determined to be 0.23 × 10$^6$ and 0.37 × 10$^6$ cells, respectively, and no significant difference was found between these BALF samples. However, the cell numbers were significantly increased to 1.22 × 10$^6$ and 5.19 × 10$^6$ in WT and *Gng13*-cKO BALF at 20 dpi, and 0.93 × 10$^6$ and 2.63 × 10$^6$ in WT and *Gng13*-cKO BALF at 25 dpi, respectively, and the latter was still significantly more than the former (*Figure 7A*). The BALF protein content also showed a similar pattern, i.e., a basal amount of protein was found in both WT and *Gng13*-cKO BALF at 0 dpi, that is, 0.36 and 0.33 mg/mL, respectively, and no significant difference was found between the two groups. At 20 dpi, however, both WT and *Gng13*-cKO BALF contained much more proteins than their corresponding samples at 0 dpi, but at 20 dpi, the *Gng13*-cKO BALF had more proteins than that of WT (2.01 vs 0.88 mg/mL), and the same is true at 25 dpi (1.65 vs 0.52 mg/mL) (*Figure 7B*). In WT, the BALF protein concentration was significantly decreased from 0.88 mg/mL at 20 dpi to 0.52 mg/mL at 25 dpi whereas the decrease of the BALF protein concentration in the *Gng13*-cKO was insignificant over the same period. The lactate dehydrogenase (LDH) activity and IL-1$\beta$ content in the BALF also showed similar patterns. No significant difference in the basal LDH activity or IL-1$\beta$ content was found between WT BALF (LDH, 33.81 U/L; IL-1$\beta$, 3.60 pg/ml) and *Gng13*-cKO BALF (LDH, 32.17 U/L; IL-1$\beta$, 3.35 pg/ml) at 0 dpi. The LDH activity then rose significantly to 73.86 U/L and 91.67 U/L at 20 dpi in WT and *Gng13*-cKO BALF, respectively, followed by a significant decrease to 53.12 U/L and 76.40 U/L, respectively, at 25 dpi. However, at both time points, the *Gng13*-cKO LDH activities were always greater than the corresponding WT activities (*Figure 7C*). WT BALF IL-1$\beta$ content also significantly increased to 38.82 pg/ml at 20 dpi, then significantly decreased to 22.03 pg/ml at 25 dpi, while the *Gng13*-cKO BALF IL-1$\beta$ levels at the two time points were 75.57 and 82.12 pg/ml, respectively, significantly higher than the corresponding WT levels. Interestingly, different from the LDH activity dynamics, the *Gng13*-cKO BALF IL-1$\beta$ level at 25 dpi did not appear to be reduced compared with that at 20 dpi (*Figure 7D*). These results indicate that while the WT injured lungs showed significant improvements in the epithelial leakage from 20 to 25 dpi, the *Gng13*-cKO injured lungs displayed higher levels of leakage and a

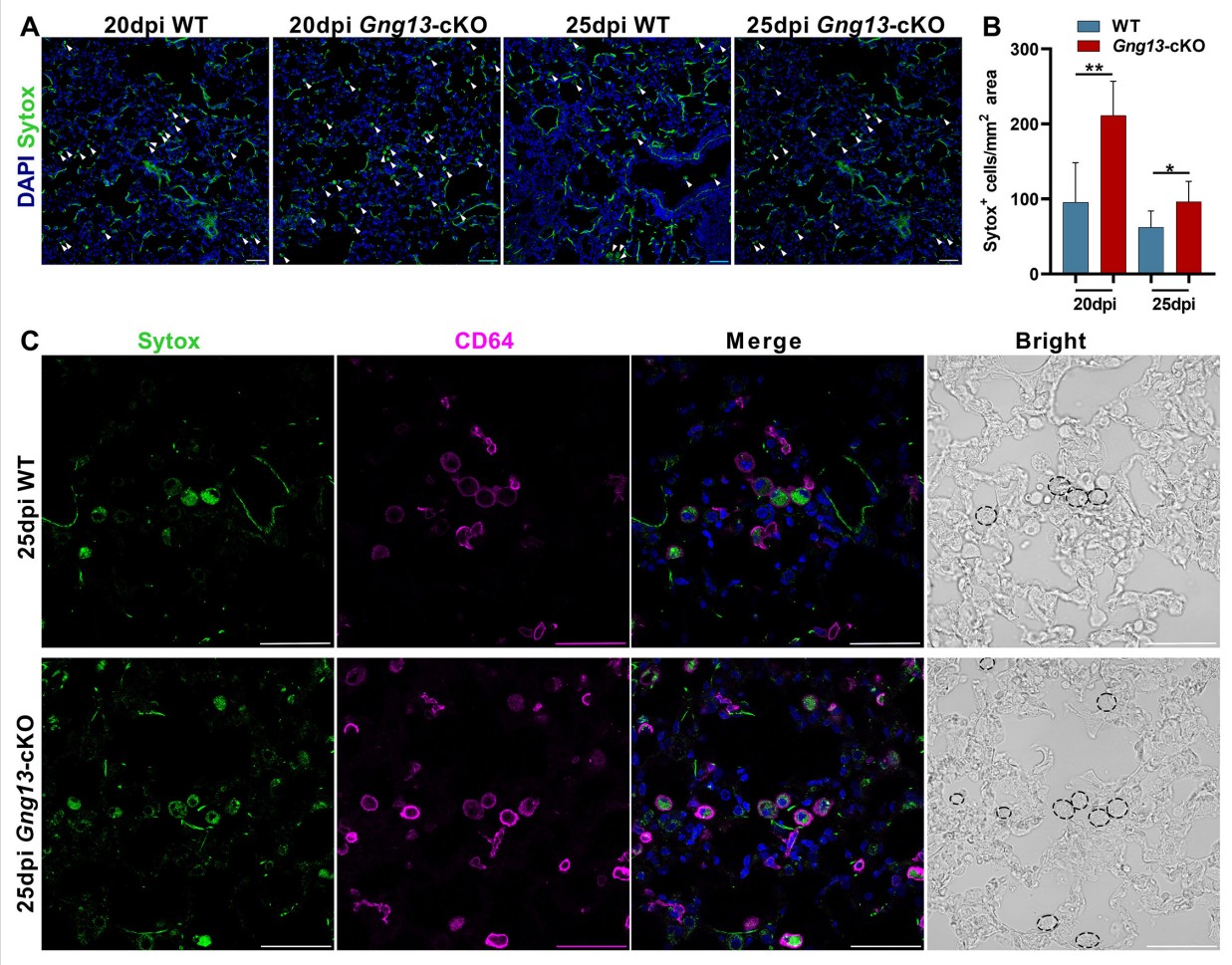

**Figure 6.** Gng13 conditional nullification leads to more pyroptotic cells in the injured lungs. (**A, B**) Histological analysis of lung tissue sections following Sytox administration reveals that significantly more Sytox⁺ cells were found in the mutant lung sections than the corresponding wild-type (WT) tissues at both 20 and 25 dpi while in both types of mice the numbers of the Sytox⁺ cells at 25 dpi seemed to be fewer than that at 20 dpi. (**C**) Immunostaining with an antibody to the macrophage marker CD64 indicates that most of the Sytox⁺ cells were macrophages in both WT and *Gng13*-cKO mice (circled by dashed lines in the bright field images on the far right). Data are presented as means ± SD (n=3), and unpaired two-tailed student t-tests were performed. Scale bars: 50 µm. *p<0.05, **p<0.01.

The online version of this article includes the following source data for figure 6:

**Source data 1.** Sytox⁺ cell densities in the injured lung tissues of wild-type (WT) and *Gng13*-cKO mice.

slower repair with a significant reduction only in the BALF LDH level, but not in the total BALF protein or IL-1$\beta$ level, from 20 to 25 dpi.

## Gng13 conditional nullification exacerbates H1N1-induced fibrosis

To assay how the *Gng13* mutation affects the recovery of the severely injured lungs, we first conducted qRT-PCR to determine the expression levels of fibrotic genes: *Col1a1*, *Fn1*, and *Timp1*. The results showed that at 0 dpi, there was no significant difference in the expression levels of these three genes between WT and *Gng13*-cKO lungs; and at 25 dpi, however, the expression levels of these genes were significantly increased in both genotypes compared with their corresponding levels at 0dpi. Comparison between the two genotypes revealed that at 25 dpi, the expression levels of all these three genes were significantly higher in the *Gng13*-cKO lungs than in WT. At 50 dpi, the expression levels of these genes showed a downward trend in both WT and *Gng13*-cKO lungs, consistent with

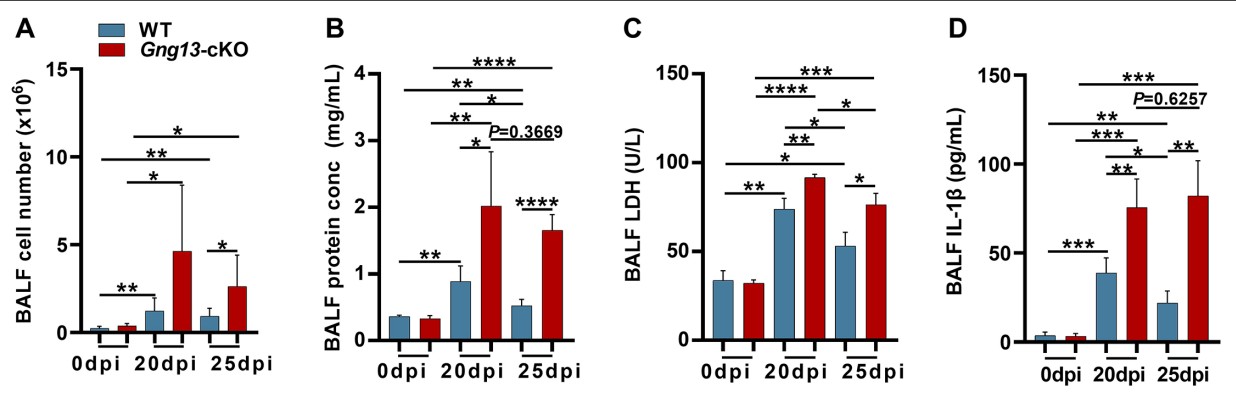

**Figure 7.** Bronchoalveolar lavage fluid (BALF) assays. (**A**) Average numbers of cells per mouse found in both WT and *Gng13*-cKO BALFs at both 20 and 25 dpi were significantly more than those at 0 dpi while those of the *Gng13*-cKO BALF were even more than those of wild-type (WT) at both 20 and 25 dpi. Data are presented as means ± SD (n=6), and unpaired two-tailed student t-tests were performed. (**B**) The protein contents of both WT BALF and *Gng13*-cKO BALF at 20 dpi and 25 dpi were significantly more than their corresponding ones at 0 dpi, but the *Gng13*-cKO BALF had significantly more proteins than WT at both 20 and 25 dpi. And in WT, the protein content was significantly reduced from 20 dpi to 25 dpi, but the reduction in the *Gng13*-cKO was insignificant. Data are presented as means ± SD (n=5), and unpaired two-tailed student t-tests were performed. (**C**) Lactate dehydrogenase (LDH) activity assays showed that at both 20 and 25 dpi, LDH activities in both WT and *Gng13*-cKO BALF were significantly higher than their corresponding ones at 0 dpi, but the *Gng13*-cKO BALF at both 20 and 25 dpi showed higher activities than the corresponding WT BALF. And both WT and *Gng13*-cKO BALF displayed significant LDH activity reduction from 20 to 25 dpi. Data are presented as means ± SD (n=3), and unpaired two-tailed student t-tests were performed. (**D**) IL-1$\beta$ ELISA assays showed a similar pattern to that of LDH, except that the reduction in the IL-1$\beta$ concentration from 20 to 25 dpi was insignificant in the *Gng13*-cKO BALF. Data are presented as means ± SD (n=4), and unpaired two-tailed student t-tests were performed. *p<0.05, **p<0.01, ***p<0.001 ****p<0.0001.

The online version of this article includes the following source data for figure 7:

**Source data 1.** Bronchoalveolar lavage fluid (BALF) assays of wild-type (WT) and *Gng13*-cKO mice.

the phase of return to homeostasis of the remodeled lungs. More specifically, the *Col1a1* expression was reduced to a similar level in WT and *Gng13*-cKO, but still significantly higher than those at 0 dpi. For *Fn1* and *Timp1*, their expression in WT at 50 dpi was much reduced, to a level similar to those at 0 dpi; and their expression in *Gng13*-cKO at 50 dpi was reduced comparing with that at 25 dpi, but still higher than that at 0 dpi or in WT at 50 dpi (*Figure 8A*).

Masson's trichrome staining was used to determine the extent of fibrosis. The results showed that 3.2% and 3.1% of fibrosis areas were found in WT and *Gng13*-cKO lungs at 0 dpi, respectively, and no significant difference was found between these two genotypes. At 25 dpi, the percentages of fibrosis areas in WT (12.3%) and *Gng13*-cKO (26.3%) were significantly more than their corresponding ones at 0 dpi. And between the two genotypes, the fibrosis in *Gng13*-cKO was even more than that in WT at 25 dpi. At 50 dpi, the fibrosis in WT (7.9%) and *Gng13*-cKO (15%) appeared to be reduced compared with those at 25 dpi, but they were still more than those at 0 dpi; and furthermore, fibrosis in *Gng13*-cKO at 50 dpi seemed to be more than that of WT at 50 dpi (*Figure 8B and C*). These results indicate that *Gng13*-cKO mice recovered not as fast as WT mice and displayed a higher levels of fibrosis.

## Discussion

Multiple subtypes of tuft cells have been found in various organs, but their distinct functions are not well understood. In this study, we used a conditional gene knockout mouse strain to nullify G$\gamma$13 expression in the ChAT-expressing tuft cells, and found that the number of ectopic tuft cells was reduced in the H1N1 virus-injured lung areas, and all of these remaining tuft cells were G$\gamma$13-negative. Furthermore, the *Gng13*-cKO mice displayed larger injured areas in the infected lungs, along with heavier immune cell infiltration, severer and prolonged inflammation, increased pyroptosis and cell death, exacerbated lung epithelial leakage and aggravated fibrosis, increased bodyweight loss and

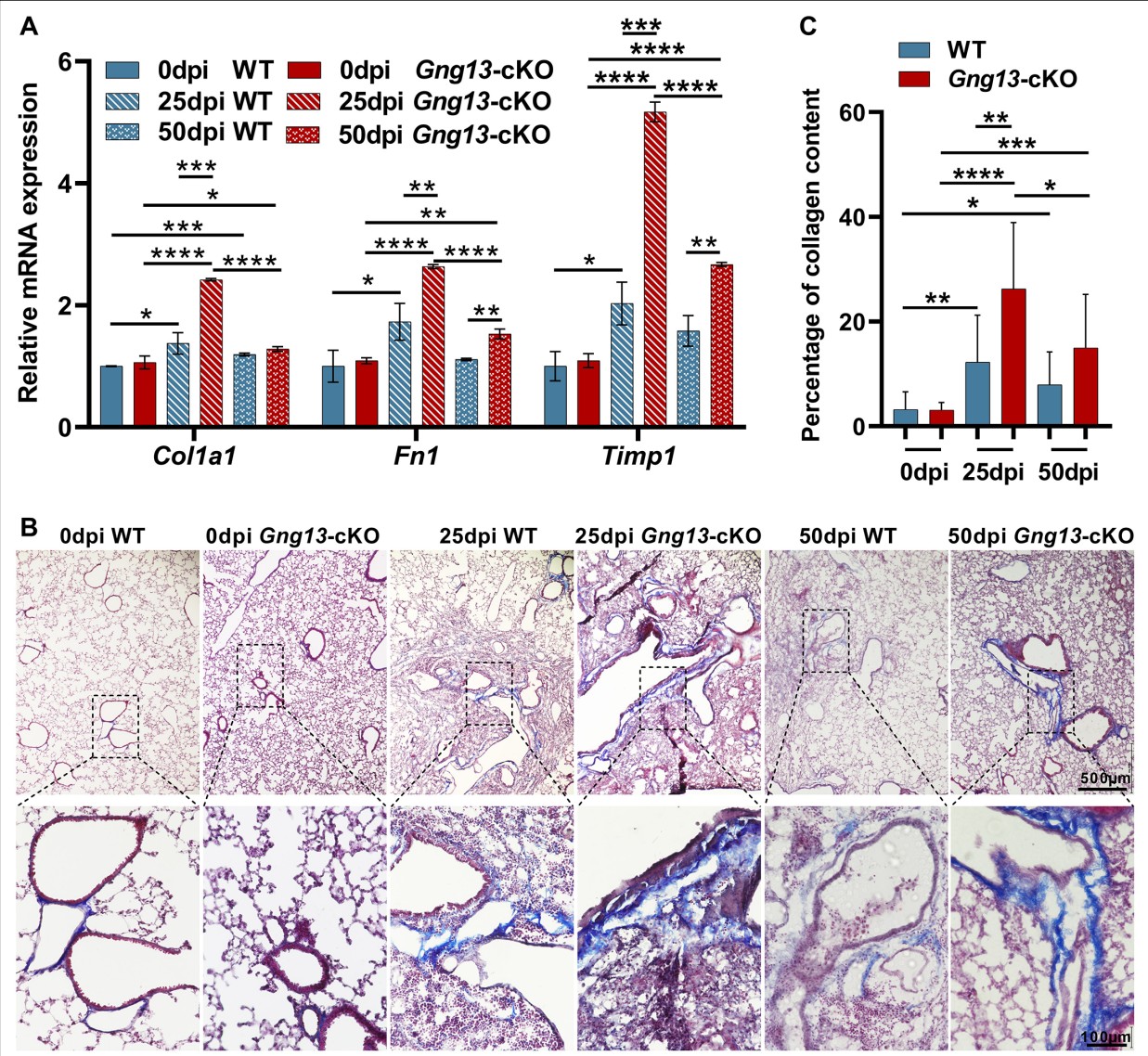

**Figure 8.** More severe fibrosis in the H1N1-infected *Gng13*-cKO lungs. (**A**) Quantitative reverse transcription-PCR (qRT-PCR) analysis of *Col1a1*, *Fn1*, and *Timp1* expression. While the expression levels of these three genes at 20 dpi in both wild-type (WT) and *Gng13*-cKO injured lungs were significantly greater than their corresponding ones at 0 dpi, the expression in *Gng13*-cKO at 25 dpi was much stronger than WT at 25 dpi. At 50 dpi, the expression levels of all these three genes were reduced compared with those at 25 dpi in both WT and *Gng13*-cKO mice, but those of *Fn1* and *Timp1* in *Gng13*-cKO mice at 50 dpi were still significantly higher than WT at 50 dpi. Data are presented as means ± SD (n=3), and unpaired two-tailed student t-tests were performed. (**B, C**) Masson's trichrome staining showed that at 25 dpi, the fibrosis percentages in WT and *Gng13*-cKO were significantly more than their corresponding ones at 0 dpi, which were reduced at 50 dpi. Data are presented as means ± SD (n=3), and unpaired two-tailed student t-tests were performed. Scale bars : 500 µm and 100 µm. *p<0.05, **p<0.01, ***p<0.001, ****p<0.0001.

The online version of this article includes the following source data for figure 8:

**Source data 1.** Fibrosis-related gene expression and percentage of collagen content in the lungs of wild-type (WT) and *Gng13*-cKO mice.

heightened fatality. These results imply that the subsets of Gγ13-positive versus negative ectopic tuft cells may play opposite but perhaps balanced roles in the inflammation resolution, tissue repair, and functional recovery, and the absence of Gγ13-positive tuft cells engenders hyper inflammation, consequently leading to aggravated outcomes.

Many signaling proteins are shared by tuft cells and taste receptor cells, including the G protein-coupled Tas1r, and Tas2r receptors (*Zhao et al., 2003*; *Max et al., 2001*; *Matsunami et al., 2000*; *Chandrashekar et al., 2000*; *Adler et al., 2000*), the heterotrimeric G protein subunits Gα-gustducin, Gα14, Gβ1, Gβ3, and Gγ13 (*Huang et al., 1999*; *Wong et al., 1996*; *Tizzano et al., 2008*),

phospholipase Cβ2 (Plcβ2), and Trpm5. Tuft cells can be deemed as solitary chemosensory cells that are dispersed over many organs of the body, monitoring internal environments, whereas taste receptor cells are aggregated into specialized end organs of taste, taste buds, located in the oral cavity of mammals, evaluating food's nutritional value or potential toxicity before ingestion (*Lindemann, 1996*; *O'Leary et al., 2019*). However, there are some important differences between taste buds and tuft cells. For example, distinct subsets of taste bud cells express Tas1rs and Tas2rs, which are employed to detect high-energy carbohydrates, nutritious amino acids, or potentially harmful and toxic substances, respectively, conveying the chemical information via different gustatory nerve fibers to the central nervous system, generating hedonic sweet and umami taste perception or unpleasant bitter taste (*Doyle et al., 2023*). In contrast, individual tuft cells can express both Tas1rs and Tas2rs, and the latter are indeed also involved in the detection of infectious microbes and invading parasites that pose risks to the host's health whereas the function of the former is less clear, and seems to regulate Tas2rs' signaling or tuft cell homeostasis (*Howitt et al., 2020*). Results of our present study indicate that Tas2rs expressed by the ectopic tuft cells are also functional, and could be involved in monitoring changes in the extracellular milieu of the injured areas during the switch from the inflammatory response to inflammation resolution, regulating disease progression.

Our results show that ectopic tuft cells first appear around day 12–14 post influenza infection and then the tuft cell number increases over the next two months, which is consistent with a previous report (*Rane et al., 2019*). However, our study further indicates that the conditional knockout of *Gng13* in the ChAT-expressing tuft cells did not affect the generation of dysplastic tuft cells up to 14 days post H1N1 infection, but did decrease ectopic tuft cells' capability to expand at 20 dpi or later, suggesting that different molecular mechanisms are employed for the initial generation versus subsequent expansion of these tuft cells: the former does not require the contribution from Gγ13-mediated signal transduction whereas the latter does require positive feedback signals released by Gγ13-mediated signaling pathways. It is still unclear how chemically or virally induced major injuries generate these dysplastic tuft cells in the distal lung tissues. Severe injuries can activate quiescent lineage-negative stem/progenitor cells in the pulmonary epithelia via a Notch signaling pathway to proliferate and differentiate into different types of pulmonary epithelial cells to repair and remodel the wounded tissues (*Rane et al., 2019*; *Vaughan et al., 2015*). But the exact signals that activate the quiescent stem cells are yet to be identified, although these signals can presumably originate from the apoptotic cells or activated immune cells following the chemical injuries or viral infection. Once activated, the stem/progenitor cells express the transcription factor Pou2f3 at a cell-type specification stage, which has been shown to be critical to the formation of both tuft cells throughout the body and Tas1r- and Tas2r-expressing receptor cells in taste buds, as nullification of the *Pou2f3* gene ablates all these taste receptor cells and tuft cells (*Gerbe et al., 2016*; *Matsumoto et al., 2011*; *Ualiyeva et al., 2020*; *Yamashita et al., 2017*). Pou2f3 may interact with other accessory proteins such as POU domain class 2 associating factors 2 and 3, i.e., Pou2af2 and Pou2af3, respectively, to fine-tune gene expression patterns and direct these cells to differentiate into subtypes of tuft cells or taste receptor cells (*Wu et al., 2022*). On the other hand, a negative regulatory mechanism has been reported, suggesting that the activating transcription factor 5 (ATF5) appears to inhibit the generation of intestinal tuft cells since more tuft cells were found in the ATF5-deficient ileum (*Nakano et al., 2023*). Presumably, some intrinsic and extrinsic factors from the severely damaged lung tissues may regulate the activation of Notch, Pou2f3, and ATF5, and together with additional transcription factors such as Gfi1b and Spdef, control the generation of subtypes of ectopic tuft cells in the injured lung.

The proliferation of intestinal tuft cells is dependent on Trpm5 (*Howitt et al., 2016*; *Lei et al., 2018*; *Luo et al., 2019*), but the tracheal tuft cells or sweet, umami, and bitter taste receptor cells in taste buds do not require *Trpm5* for their expansion (*Damak et al., 2006*; *Zhang et al., 2003*; *Finger et al., 2005*; *Hollenhorst et al., 2022*). We and others have found that *Trpm5* is not required for the proliferation of the dysplastic pulmonary tuft cells (*Barr et al., 2022*; *Huang, 2022*), but conditional nullification of *Gng13* ablated part of these ectopic tuft cells and reduced the total number of these cells, indicating that different tuft cells utilize different mechanisms to control their proliferation. In the small intestine, the hyperplasia of tuft cells as well as goblet cells is regulated by the Trpm5-IL25-ILC2-IL13 molecular circuit. Pathogens or microbial metabolites activate the succinate receptor Sucnr1, Tas2rs, Vmn2r26, and prostaglandin receptors and possibly other GPCRs, which then stimulate the heterotrimeric G proteins composed of Gα-gustducin, Gβ1, Gβ3, and Gγ13. Single-cell RNAseq

data have shown that many of these GPCRs and G protein subunits are co-expressed in the intestinal as well as ectopic pulmonary tuft cells although additional G protein subunits could also be involved in the signal transduction (*Barr et al., 2022*). Upon activation, the heterotrimeric G$\alpha\beta\gamma$ dissociates into the G$\alpha$ and G$\beta\gamma$13 moieties, each of which can act on their own effectors. While G$\alpha$-gustducin's effectors are still to be determined, G$\beta\gamma$13 can act on the phospholipase Plc$\beta$2, or other effectors such as adenylate cyclases, phospholipases, receptor kinases, voltage-gated ion channels, and inward rectifying ion channels (*Campbell and Smrcka, 2018*; *Kankanamge et al., 2022*). Among them, Plc$\beta$2 can generate the second messengers diacylglycerol (DAG) and inositol trisphosphate (IP3) (*Huang et al., 1999*). DAG can activate a number of protein kinases, but their eventual physiological effect is yet to be elucidated. On the other hand, IP3 can bind to and open its channel receptor IP3R on the endoplasmic reticulum (ER), releasing the calcium ions from the ER and consequently increasing the cytosolic calcium concentration, leading to the opening of Trpm5, a calcium-activated non-selective monovalent cation channel, depolarizing the membrane potential and releasing certain output signals (*Zhang et al., 2007*; *Liman, 2014*). The release of the cytokine IL-25, acetylcholine (ACh), ATP, calcitonin gene-related peptide (CGRP) and component C3 from tuft cells and taste bud cells is dependent on Trpm5, but that of some antimicrobial substances and eicosanoids do not require Trpm5 (*Lee et al., 2014*), indicating that Trpm5 regulates only some output signals' release. Since the expansion of ectopic tuft cells is independent of Trpm5, we believe that Trpm5-dependent output signaling molecules may not be involved in the regulation of these tuft cells' proliferation. On the other hand, the elimination of G$\gamma$13-expressing pulmonary tuft cells in the *Gng13*-cKO mice suggests that the effectors activated by the G$\beta\gamma$13 moiety are involved in the regulation of output signals that promote the proliferation of G$\gamma$13-expressing tuft cells. The differential impact of *Trpm5* knockout versus *Gng13* abolishment on tuft cell proliferation is probably attributed to the fact that G$\gamma$13 is upstream of Trpm5 in the signaling cascades, thus regulating more signaling pathways, and when mutated, having a more profound effects than Trpm5. Further investigation is needed to determine what signals the G$\gamma$13-positive and -negative ectopic pulmonary tuft cells are needed to expand, and with this knowledge, one could intentionally manipulate increasing the clinically beneficial subtypes while decreasing the harmful ones of dysplastic tuft cells.

Previous reports indicate that the elimination of all ectopic tuft cells by knocking out the *Pou2f3* gene did not significantly affect the pathogenesis and outcome of H1N1-induced severe injury (*Barr et al., 2022*; *Huang, 2022*). In our study, elimination of G$\gamma$13-expressing ectopic pulmonary tuft cells rendered severer symptoms following H1N1 infection, ranging from more immune cell infiltration, increased pyroptotic gene expression and cell death, larger injury areas in the lungs, more BALF protein contents and immune cells, and larger areas of fibrosis, to a slower recovery processes and a higher fatality rate. The striking discrepancy may be caused by the imbalance of tuft cells' diverse functions in the *Gng13*-cKO lung. Since the ectopic tuft cells are absent until 12–14 dpi, these cells may not be involved in the initial inflammatory responses to viral infection or the subsequent viral clearance. When the pathogeneses enter the later stages, the *Gng13*-cKO lungs have a much reduced number of tuft cells with only a few of G$\gamma$13-negative tuft cells remaining, leading to much severer symptoms, indicating that the missing tuft cells may play a major role in the transition from the inflammatory responses to inflammation resolution and subsequent tissue repair processes while the remaining tuft cells are unable to effectively contribute to these processes, or even worse, stimulate inflammation.

Indeed, some tuft cells have been shown to produce pro-inflammatory cytokines and lipid mediators such as IL-25, prostaglandins, and leukotrienes that activate innate and adaptive immune responses (*Kotas et al., 2022*; *Ualiyeva et al., 2021*; *Hollenhorst and Krasteva-Christ, 2023*), and single cell RNAseq analyses indicate that pulmonary ectopic tuft cells express genes that encode enzymes producing specialized pro-resolving mediators (SPMs), for example, *Alox5*, *Alox12e*, *Alox15*, *Ptgs2*, encoding arachidonate 5-lipoxygenase (5-LOX), arachidonate 12-lipoxygenase (12-LOX), arachidonate 15-lipoxygenase (15-LOX), prostaglandin-endoperoxide synthase II (also known as cyclooxygenase II, COX-2), respectively, as well as the genes encoding various isoforms of cytochrome P450: *Cyp51*, *Cyp2f2*, *Cyp7b1*, and *Cyp2j6* (*Barr et al., 2022*). These enzymes can utilize membrane lipids such as docosahexaenoid acid, eicosapentaenoid acid and arachidonic acid to biosynthesize SPMs, including lipoxins, resolvins, protectins, and maresins, which can limit further infiltration of neutrophils, recruit and stimulate macrophage phagocytosis to remove dead cells and cell debris, suppress the production of pro-inflammatory mediators and cytokines, and increase the production of anti-inflammatory

mediators and cytokines, reduce fibrosis and accelerate tissue repair and remodeling (*Serhan et al., 2020*; *Panigrahy et al., 2021*; *Serhan et al., 2018*). So far, the SPMs maresin 1 and resolvin D1 have been shown to inhibit virus- and bacterium-induced inflammation in the lung, respectively (*Krishnamoorthy et al., 2023*; *Codagnone et al., 2018*). Together, our data from this study implies that nullification of *Gng13* not only significantly reduces the number of ectopic tuft cells but also disrupts the switch from producing proinflammatory lipid mediators to producing anti-inflammatory SPMs. Given the heterogeneity of tuft cells (*Xiong et al., 2022*; *Bankova et al., 2018b*; *Haber et al., 2017*), we hypothesize that the remaining dysplastic tuft cells may continue to produce proinflammatory cytokines and lipid mediators whereas the anti-inflammatory agents-producing tuft cells are absent in the *Gng13*-cKO mice, consequently leading to severer outcomes.

In conclusion, to our knowledge, it is the first time to postulate that different subtypes of tuft cells may play different subtype-specific roles in the phases of infection, response, resolution, and recovery upon H1N1 infection or other severe injuries. Our data reveal that G$\gamma$13-expressing ectopic tuft cells promote the inflammation resolution and recovery while other dysplastic lung tuft cells contribute to extended inflammation and perhaps immune storms as well.

# Materials and methods

## Key resources table

| Reagent type (species) or resource | Designation | Source or reference | Identifiers | Additional information |
|---|---|---|---|---|
| Strain, strain background (*Mus musculus*) | *Chat*-IRES-Cre | The Jackson Laboratory | Cat#: 006410 | |
| Strain, strain background (*Mus musculus*) | *Trpm5*$^{-/-}$ | The Jackson Laboratory | Cat#: 005848 | |
| Strain, strain background (*Mus musculus*) | *Gng13*$^{flox/flox}$ | *Li et al., 2013* | N/A | |
| Strain, strain background (*Mus musculus*) | *Gng13*-cKO | The Jackson Laboratory; *Li et al., 2013* | Cat#006410 | Strain: *Chat*-IRES-Cre × *Gng13*$^{flox/flox}$ |
| Antibody | Anti-Dclk1 (rabbit polyclonal) | Abcam | ab31704; RRID:AB_873537 | IF (1:1,000) |
| Antibody | Anti-Dclk1-488 (rabbit monoclonal) | Abcam | ab202754 | IF (1:1,000) |
| Antibody | Anti-G$\alpha$gust (rabbit polyclonal) | Santa Cruz Biotechnology | sc-395 | IF (1:500) |
| Antibody | Anti-Plc$\beta$2 (rabbit polyclonal) | Santa Cruz Biotechnology | sc-206 | IF (1:500) |
| Antibody | Anti-CD45 (rabbit polyclonal) | Abcam | ab10558; RRID:AB_442810 | IF (1:500) |
| Antibody | Anti-CD64 (rabbit monoclonal) | Thermo Fisher Scientific | MA5-29706; RRID:AB_2785530 | IF (1:200) |
| Antibody | Anti-Gsdmd (rabbit polyclonal) | Affinity | AF4012; RRID:AB_2846776 | IF (1:400) |
| Antibody | Anti-Gsdme (rabbit polyclonal) | Proteintech | 13075-1-AP; RRID:AB_2093053 | IF (1:500) |
| Antibody | Anti-F4/80-FITC (rabbit monoclonal) | Thermo Fisher Scientific | 11-4801-81; RRID:AB_2735037 | IF (1:500) |
| Antibody | Anti-EpCAM (rabbit monoclonal) | Thermo Fisher Scientific | 11-5791-82; RRID:AB_11151709 | IF (1:500) |
| Antibody | Anti-Krt5 (rabbit polyclonal) | BioLegend | 905504; RRID:AB_2734679 | IF (1:500) |
| Antibody | Anti-Trpm5 (rabbit polyclonal) | Sigma-aldrich | AV35242; RRID:AB_1858368 | IF (1:500) |
| Antibody | Anti-Caspase-3 (rabbit monoclonal) | Abcam | ab32351 | IF (1:500) |

*Continued on next page*

*Continued*

| Reagent type (species) or resource | Designation | Source or reference | Identifiers | Additional information |
|---|---|---|---|---|
| Antibody | Anti-Gγ13 (rabbit polyclonal) | *Huang et al., 1999* | N/A | IF (1:200) |
| Antibody | Donkey anti-rabbit IgG H&L (Alexa Fluor488) | Abcam | ab150073; RRID:AB_2636877 | Secondary antibody IF (1:500) |
| Antibody | Donkey anti-rabbitIgG H&L (Alexa Fluor568) | Abcam | ab175470; RRID:AB_2783823 | Secondary antibody IF (1:500) |
| Antibody | Anti-NLRP3 (rabbit monoclonal) | Abcam | ab263899; RRID:AB_2889890 | WB (1:1,000) |
| Antibody | Anti-Gsdmd (rabbit monoclonal) | Abcam | ab219800; RRID:AB_2888940 | WB (1:1,000) |
| Antibody | Anti-caspase-1 (mouse monoclonal) | Santa Cruz Biotechnology | sc-56036; RRID:AB_781816 | WB (1:1,000) |
| Antibody | Anti-IL-1 (goat polychonal) | R&D Systems | AF-401-SP | WB (1:800) |
| Antibody | Anti-β-actin (mouse monoclonal) | Beyotime | AA128-1 | WB (1:1,000) |
| Antibody | Anti-rabbit (goat polychonal) | Beyotime | A0208 | HRP-conjugated secondary antibody WB (1:2,000) |
| Antibody | Anti-goat (donkey polychonal) | Beyotime | A0181 | HRP-conjugated secondary antibody WB (1:2,000) |
| Antibody | Anti-mouse (goat polychonal) | Beyotime | A0216 | HRP-conjugated secondary antibody WB (1:2,000) |
| Sequence-based reagent | PCR/ sequencing primers | This paper | N/A | *Table 1* |
| Commercial assay or kit | LDH-GloTM Cytotoxicity Assay | Promega | Cat#: J2380 | |
| Commercial assay or kit | Mouse IL-1 beta DuoSet ELISA | R&D Systems | Cat#: DY401 | |
| Commercial assay or kit | Modified Masson's Trichrome Stain Kit | absin | Cat#: abs9348 | |
| Chemical compound, drug | Bleomycin | Selleck | Cat#: s1214 | 6 mg/kg bodyweight |
| Chemical compound, drug | House dust mite extract | Biolead | Cat#: XPb70D3A2.5 | 2 mg/kg bodyweight |
| Chemical compound, drug | Lipopolysaccharide | Sigma | Cat#: L2630 | 1 mg/kg bodyweight |
| Chemical compound, drug | Denatonium benzoate | Sigma | Cat#: D5765 | 1 mM |
| Chemical compound, drug | Quinine | MedChemExpress | Cat#: HY-D0143 | 100 μM |
| Chemical compound, drug | Allylisothiocyanate | Sigma | Cat#: 36682 | 3 mM |
| Chemical compound, drug | Gallein | APExBIO | Cat#: B7271 | 100 μM |
| Chemical compound, drug | U73122 | Sigma | Cat#: U6756 | 10 μM |
| Chemical compound, drug | 1× Sytox Green | Beyotime | Cat#: C1070S | 1× |
| Software, algorithm | Graphpad Prism 9 | Graphpad | RRID:SCR_002798 | https://www.graphpad-prism.cn/?c=i&a=prism |

## Experimental design

This study was designed to investigate the possible roles of the taste signaling proteins, the heterotrimeric G protein subunit Gγ13 and the transient receptor potential ion channel Trpm5, in the host response to H1N1 influenza virus infection as well as in the subsequent inflammation resolution, tissue remodeling, and recovery using transgenic and conditional gene knockout mouse models with wild-type (WT) mice as control. Animals were intranasally inoculated with H1N1 viruses at a sublethal dosage, their bodyweights were measured, mortality rates determined; the lung tissues and bronchoalveolar lavage fluid were molecularly, biochemically, immunologically, and statistically analyzed.

## Animals

C57BL/6 mice were purchased from the Shanghai SLAC Laboratory animal company. *Chat*-IRES-Cre, *Trpm5*-knockout/lac Z knockin (i.e., *Trpm5*$^{-/-}$), and *Ai9* (Jax stock numbers 006410, 005848 and 7909) were obtained from the Jackson laboratory. The *Gng13*$^{flox/flox}$ mice were generated previously (*Li et al., 2013*) and bred with the *Chat*-IRES-Cre mice to conditionally knock out the *Gng13* gene in the choline acetyl-transferase (ChAT)-expressing cells of the *Gng13*-cKO mutant mice. Mice were bred to more than 98% of C57BL/6 genetic background and maintained under a 12-hr light/dark cycle with access to water and food ad libitum in the Laboratory Animal Center of Zhejiang University. Progeny was genotyped by PCR, and both male and female mice at 6 to 8 weeks old were used in the experiments. Studies involving animals were approved by the Zhejiang University Institutional Animal Care and Use Committee, and performed following the NIH 'Guidelines for the Care and Use of Laboratory Animals'.

## Animal models of virally or chemically induced lung injury

Mice were anesthetized with 5% chloral hydrate (Sangon Biotech, A600288), and then intranasally administrated with 120 pfu of influenza virus A/Puerto Rico/8/1934 in 25 µl PBS at 0 day post infection (dpi). Control mice were intranasally instilled with 25 µl PBS only. The mice were checked every day and weighted every two days and euthanized for analysis at specific time points.

For chemically injured mouse models, mice were intraperitoneally administrated with a single dose of bleomycin (Selleck, s1214) at 6 mg/kg bodyweight, or first anesthetized and then injected into the trachea with the house dust mite extract (Biolead, XPb70D3A2.5) at 2 mg/kg bodyweight, one dose per day for three consecutive days, or injected through the trachea a single dose of lipopolysaccharide (LPS) (Sigma, L2630) at 1 mg/kg bodyweight. The lung tissues were dissected out for studies 14 days after the first injection.

## Fluorescence-activated cell sorting

The heterogeneous *Chat*-Cre: Ai9 mice at 25 dpi were anesthetized and transcardially perfused with cold PBS; their lungs were then isolated and dissociated into single cells as described previously (*Zhao et al., 2020*; *Barkauskas et al., 2013*). Briefly, the lung tissues were minced into pieces of 1 mm$^3$ by a pair of ophthalmic scissors, which were then digested with 1 mg/mL dispase II (Sigma, D4693), 3 mg/mL collagenase I (Sigma, C0130), and 0.5 mg/mL DNase I (Sigma, DN25) in DMEM/F12 medium (Thermo Fisher Scientific, C11330500BT) with 1% P/S for 50 min at 37°C on a shaker. The cell suspension was then filtered with a 70-µm cell strainer (Biosharp, BS-70-XBS) and treated by red blood cell lysis buffer (Beyotime, C3702) for 3 min at RT. The cell suspension was then again filtered by a 40-µm cell strainer (Biosharp, BS-40-XBS), and the cells were collected and resuspended in the collection buffer (DPBS plus 0.04% BSA) (*Figure 2—figure supplement 2*). Cell sorting and data analyses were performed on a Beckman Coulter FACS flow cytometer.

## Immunohistochemistry

Lungs were harvested and processed as described previously (*Zhao et al., 2020*). Briefly, freshly dissected mouse lungs were fixed with 4% paraformaldehyde (PFA) at RT for 2.5 hr on a shaker, followed by cryoprotection in 30% sucrose in PBS at 4°C overnight. The lungs were incubated in 30% sucrose in PBS pre-mixed with an equal volume of optimal cutting temperature compound (OCT, Tissue-Tek) at RT for 2 hr on a shaker, and then embedded in 100% OCT and placed in a −80°C freezer for 2 hr, which were then sliced into 12 µm-thick sections on a cryostat (Thermo scientific, HM525). The tissue sections were blocked in the blocking buffer (PBS plus 3% BSA, 0.3% Triton X-100, 2% donkey serum, and 0.1% sodium azide) for 1 hr at RT, followed by incubation at 4°C overnight with primary antibodies diluted in the blocking buffer (rabbit anti-Dclk1 1:1000, Abcam, ab31704; rabbit anti-Dclk1-488 1:1000, Abcam, ab202754; rabbit anti-Gαgust 1:500, Santa Cruz Biotechnology, sc-395; rabbit anti-Plc$\beta$2 1:500, Santa Cruz Biotechnology, sc-206; rabbit anti-CD45 1:500, Abcam, ab10558; rabbit anti-CD64 1:200, Thermo Fisher Scientific, MA5-29706; rabbit anti-Gsdmd 1:400, Affinity, AF4012; rabbit anti-Gsdme 1:500, Proteintech, 13075-1-AP; rabbit anti-F4/80-FITC 1:500, Thermo Fisher Scientific, 11-4801-81; rabbit anti-EpCAM 1:500, Thermo Fisher Scientific, 11-5791-82; rabbit anti-krt5 1:500, BioLegend, 905504). The slides were washed and incubated with the secondary antibodies: donkey anti-rabbit IgG H&L (Alexa Fluor 488) 1:500, Abcam, ab150073;

donkey anti-rabbit IgG H&L (Alexa Fluor 568) 1:500, Abcam, ab175470, for 1.5 hr at RT. Fluorescent images were captured using an FV3000 laser scanning confocal microscope (Olympus) and VS200 slide scanner (Olympus).

## Calcium imaging

TdTomato$^+$ cells were isolated and FACS-sorted as described above from the H1N1-infected *Chat*-Cre: Ai9 mice. These cells were seeded on laminin-coated chambers and cultured in the imaging buffer (1X HBSS, 10 mM HEPES, 1 mM sodium pyruvate) for 30 min. Cells were loaded with 5 μM fluorescent Ca$^{2+}$ indicator Fluo-4 AM (Dojindo, F312) and Pluronic F-127 for 40 min at 37 °C, 5% CO$_2$, following the previously reported protocol (*Luo et al., 2019*). The cells were gently washed twice with HBSS to remove any excess Fluo-4 AM and Pluronic F-127. Denatonium benzoate (D.B., Sigma, D5765) and quinine (MedChemExpress, HY-D0143) were used as bitter substances. Allyl isothiocyanate (AITC, Sigma, 36682), gallein (APExBIO, B7271), and U73122 (Sigma, U6756) were used as inhibitors for bitter taste receptor, G protein $\beta\gamma$ subunit, and Plc$\beta$2 activity, respectively. The cells were first stimulated by the bitter compounds of 1 mM D.B. or 100 μM quinine, then blocked by incubating with 3 mM AITC, 100 μM gallein or 10 μM U73122 for 10 min before the inhibitors were removed; and D.B. or quinine applied again to assess the cells' calcium responsiveness. Images were captured with an excitation wavelength of 494 nm and emission wavelength of 516 nm using an IX83 total internal reflection fluorescence microscope (Olympus).

## Bronchoalveolar lavage fluid (BALF) assays

BALF were prepared as previously described (*Akbari et al., 2003*). Briefly, after the trachea was exposed, 0.5 mL ice-cold PBS was gently instilled into the lung and then all the fluid was collected, which was repeated three times. A total of 1.5 mL BALF was collected, which was centrifuged at 300 × g for 10 min at 4°C. The supernatants were used to determine lactate dehydrogenase activity and IL-1$\beta$ level using LDH-Glo Cytotoxicity Assay (Promega, J2380) and Mouse IL-1 beta DuoSet ELISA (R&D Systems, DY401) following the manufacturer's instruction. The cells in the pellet were resuspended, a portion of which was used to count cells whereas the rest was used to extract proteins using RIPA buffer (Sangon Biotech, C500005); and the protein concentrations were determined by a BCA protein assay kit (Sangon Biotech, C503021) following the manufacturer's instruction.

## Lung wet/dry weight (W/D) ratio

Mice were sacrificed with a lethal dose of 5% chloral hydrate. The lungs were isolated and cleaned to remove any adherent blood. The wet weight of lungs was immediately determined, followed by drying in a heated stove at 65°C for 48 hr. The W/D ratio was calculated by dividing the wet lung weight by its corresponding dry weight.

## Quantitative reverse transcription-PCR

Lungs from H1N1-infected and -uninfected mice were dissected out and washed three times with ice-cold PBS. The damaged lung areas of the infected mice and the corresponding lung areas of the uninfected control mice were excised to isolate total RNAs using TaKaRa MiniBEST Universal RNA Extraction Kit (TaKaRa, 9767), which were reverse transcribed into cDNAs using the RevertAid First Strand cDNA Synthesis Kit (Thermo Fisher Scientific, K1622). qPCR reactions were set up using iQ SYBR Green Supermix (Bio-rad, 1708884), and run on the CFX Connect Real-Time System. Relative expressions were calculated and normalized to the expression of an internal control gene *Actb*. The primer pairs used in these qPCR reactions are listed in *Table 1*.

## Western blotting

Lung lobes from H1N1-infected and -uninfected control mice were harvested. The damaged lung tissues from the infected mice and the tissues from the corresponding areas of the control lungs were collected to extract proteins using RIPA buffer (Sangon Biotech, C500005), which were centrifuged at 14,000× g for 25 min to remove any debris. The protein concentrations in the supernatants were determined using a BCA protein assay kit (Sangon Biotech, C503021), and an equal amount of protein, i.e., 20 or 40 μg from each sample was loaded onto a 10-12% SDS-PAGE gel for protein separation. The proteins on the gel were then transferred to PVDF membranes (Millipore, IPVH00010),

**Table 1.** Sequences of primers used for qPCR.

| Species | Gene name | Sequences (5'-3') | Sequences (5'-3') |
|---|---|---|---|
| Mouse | Tas2r102 | F:CTCCTGCTAATCTTCTCTTTGTG | R:GGGTCTCTGTGTCTTCTGG |
| | Tas2r103 | F:AGCACAGTGGCCCACATAAA | R:TGGCCTGTGGGAAAAGCTAC |
| | Tas2r104 | F:GCAACACATCCTGGCTGAT | R:CCCCATATTGGCAAAAACAT |
| | Tas2r105 | F:CAGAAGGCATCCTCCTTTCCA | R:GCCCAGTCCATGCAGTTTAC |
| | Tas2r106 | F:AGCCACATTCTTCTCAACCT | R:AGCATGTAATGATAGCCACCA |
| | Tas2r107 | F:CTGGTTTGACAGCCACATGC | R:TGCCTTCAAAGAGGCTTGCT |
| | Tas2r108 | F:GTTTCTCCTGTTGAAACGGACT | R:GTGAGGGCTGAAATCAGAAGA |
| | Tas2r109 | F:GTCAAATTCAGGTGTTAGGAAGTC | R:CACAGGGAGAAGATGAGCAG |
| | Tas2r110 | F:AGGTCAATGCCAAACCACCT | R:CAGCAGGAAGGAGAACCCTG |
| | Tas2r113 | F:AGAATATGCAGCACACCGCC | R:CAATGATGGTTTGCAGGGCTC |
| | Tas2r114 | F:ACACATCTTGGCAGATCCACA | R:TTTGATTCCATCTGCCTGCGA |
| | Tas2r115 | F:CCTTTGGTGTATCCTTGATAGCTT | R:CTGCATCTTCCTTACATGTTTCA |
| | Tas2r116 | F:AAGGTTTGGAGTGCTCTGCT | R:AGCTGTTCTTGCAACCTGTGT |
| | Tas2r117 | F:CCCTGTGGACACATCACAAG | R:TCACAGTTTGTAGGGCTTTGAA |
| | Tas2r118 | F:CACTGGGTGCAGATGAAACA | R:CTTCAGAACAGTGAACTGAGCTTT |
| | Tas2r119 | F:TGCACAGCTGGGTCTATCCT | R:CACCAAGCCATGTGGCAAAC |
| | Tas2r120 | F:TTGGTTTGTTGTGGGCAATGT | R:TCAGTATGGTCCCCAGCCAA |
| | Tas2r121 | F:CTGGTCTTATTGGAGATGATTGTG | R:GGAGAAGATTAACAGGATGAAGGA |
| | Tas2r122 | F:GCTTATTGTGGCAAGCTCCA | R:AACCTCCACAATGACACACCA |
| | Tas2r123 | F:TTCATGCTGTGCCCACATTT | R:AAGACACCGAGGCATGTAGT |
| | Tas2r124 | F:CTACGGCCCACAGAAATGCC | R:AGCTGCCTCATTACCCAAAGA |
| | Tas2r125 | F:CTAAAGGCTCCGAAGACACCA | R:AACAGGAGAAAGGCCACTACC |
| | Tas2r126 | F:GTGTGTGGGATTGGTCAACA | R:GCTCCCGGAGTACTCAACC |
| | Tas2r129 | F:TTTAGCATGTGGCTTGCTGC | R:AGAGGCCCAAAGACATGAGC |
| | Tas2r130 | F:TGCATTCATTGCACTGGTAAA | R:GATTAAATCAATAGAGGCAATCTTCC |
| | Tas2r131 | F:TAGCCCACATTTCCCATCC | R:CAAGCACACCTCTCAATCTCC |
| | Tas2r134 | F:GCCTGGGAAGTGGTAACCTA | R:GTTGCTTAGTATCAGAATGGTGGA |
| | Tas2r135 | F:CCATCATGTCCACAGGAGAA | R:TCAGTAGTCTGACATCCAAGAACTGT |
| | Tas2r136 | F:GCAAAGAGCTTTCTCAAAGACC | R:AGGGATAGATAAACAGGGAAACACT |
| | Tas2r137 | F:CTGGCTCAAATGGAGAGCTT | R:GGTACTGACACAGGATAAGAGCAG |
| | Tas2r138 | F:CAAACCAAGTGAGCCTCTGG | R:GAGAAGCGGACAATCTTGGA |
| | Tas2r139 | F:ATGGCTCAACCCAGCAACTAC | R:ACAGCCATGACAATCCCACT |
| | Tas2r140 | F:GAAGAACATGCAACACAATGC | R:AGGGCCTTAATATGGGCTGT |
| | Tas2r143 | F:CATTGGCCTCTATGTTGCAG | R:TGTCCGGTTCCTCATCCA |
| | Tas2r144 | F:AAGCAGAAAATCATAGGGCTGA | R:TGAAGGAAACCAACACTGACA |
| | Gsdma | F:CTAGTCTGATCCTTCCCATGTGT | R:CAGTGTGGGCAGTAACGTGT |
| | Gsdma2 | F:CCCTTCCCTGGAAAATCTGGA | R:CCGGGTGACATCCTCAAACA |
| | Gsdma3 | F:ACTGAAATGCCTGCTCATCTT | R:CATCAGGAGATGGGCTTAGTGG |
| | Gsdmc | F:TCTTCCCGGTTGGCTTTGAAA | R:AGGACTTAACAAACCCTGCTTC |

*Table 1 continued on next page*

*Table 1 continued*

| Species | Gene name | Sequences (5'-3') | Sequences (5'-3') |
|---------|-----------|-------------------|-------------------|
| | Gsdmc2 | F:CTGTGGAATGCTTGTCCGATG | R:CCTCCAGGTCCGTTGATTGG |
| | Gsdmc3 | F:AGCCCGCCCATCTAGATTTC | R:TGCCCCAACTGACTCAACTC |
| | Gsdmc4 | F:TGAGGAGCCTGCCAATCTAAA | R:ATGTGGGGTGCTAGAATCCTT |
| | Gsdmd | F:GATCAAGGAGGTAAGCGGCA | R:CACTCCGGTTCTGGTTCTGG |
| | Gsdme | F:GTCAGCAGAGGCAAACAATCG | R:TTTCTTCGCTGTGCTGCTTG |
| | Tlr4 | F:GTTCTCTCATGGCCTCCACT | R:AGGGACTTTGCTGAGTTTCTGAT |
| | Nlrp3 | F:CCCCTTTATTTGTACCCAAGGCT | R:GCAACGGACACTCGTCATCT |
| | Asc | F:GACAGTACCAGGCAGTTCGT | R:AGTCCTTGCAGGTCAGGTTC |
| | Casp1 | F:CCGCGGTTGAATCCTTTTCAG | R:TGTGCGCATGTTTCTTTCCC |
| | Il1b | F:TGCCACCTTTTGACAGTGATG | R:AAGGTCCACGGGAAAGACAC |
| | Col1a1 | F:AGAGCGGAGAGTACTGGATCG | R:TCGAACGGGAATCCATCGGT |
| | Fn1 | F:AACAGAAATTGACAAGCCGTC | R:TCTGTTTGATCTGGACTGGCA |
| | Timp1 | F:GCAAAGAGCTTTCTCAAAGACC | R:AGGGATAGATAAACAGGGAAACACT |

which were blocked in 5% BSA for 1 hr at RT. The membranes then were incubated at 4°C overnight with the primary antibodies diluted in the blocking buffer (rabbit anti-NLRP3 1:1000, Abcam, ab263899; rabbit anti-Gsdmd 1:1000, Abcam, ab219800; mouse anti-caspase-1 1:1000, Santa Cruz Biotechnology, sc-56036; goat anti-IL-1$\beta$ 1:800, R&D Systems, AF-401-SP; mouse anti-$\beta$-actin 1:1000, Beyotime, AA128-1), followed by incubation with the HRP-conjugated secondary antibodies (goat anti-rabbit 1:2000, Beyotime, A0208; donkey anti-goat 1:2000, Beyotime, A0181; goat anti-mouse 1:2000, Beyotime, A0216) for 1.5 hr at RT. Protein bands on the membranes were visualized using a chemiluminescent imaging system (Tanon, 5200) and quantified using the ImageJ software.

## Collagen deposition assay

Lung tissue sections were analyzed with hematoxylin and eosin (H&E) staining and Masson's trichrome staining for the assessment of the proportion of collagen content (*Jia et al., 2019*). Modified Masson's Trichrome Stain Kit (absin, abs9348) was used according to the manufacturer's instructions. The sections then were photographed using a multizoom AZ100 microscope (Nikon) and the Masson's trichrome-stained collagen areas were determined using the ImageJ software, divided by the total area of the section. For each treatment, three or more mice were used.

## Sytox green staining

Each mouse was intranasally administered with 1 mL 1x Sytox Green (Beyotime, C1070S) (*Halverson et al., 2015*). Ten minutes later, the lungs were harvested and cut into 12 μm-thick sections as previously described (*Zhao et al., 2020*). Green fluorescence (Ex. 490; Em. 535) was measured using FV3000 laser scanning confocal microscope (Olympus).

## Statistical analysis

Experimental data were obtained from three or more biological replicates, presented as mean ± SD using Graph Prism 9 software. Unpaired two-tailed Student's *t*-tests were performed and p-values <0.05 were considered statistically significant. Animal survival rates were determined using the Kaplan-Meier method and data were analyzed with the log-rank test.

## Additional files

This PDF file includes: *Table 1*, *Figure 1—figure supplement 1* to *Figure 5—figure supplement 2*.

## Acknowledgements

We thank all Huang lab members for their helpful discussion and support. This research was in-part supported by the Starry Night Science Fund of Zhejiang University Shanghai Institute for Advanced Study Grant SN-ZJU-SIAS-005 (to LHuang) and SN-ZJU-SIAS-003 (to RZ) National Key Research and Development Program of China Grant 2021YFF1200803 (to LHuang), 2021YFF1200404 and 2021YFA1201200 (to RZ) National Natural Science Foundation of China U1967217 (to RZ) National Center of Technology Innovation for Biopharmaceuticals NCTIB2022HS02010 (to RZ) National Independent Innovation Demonstration Zone Shanghai Zhangjiang Major Projects ZJZX2020014 (to RZ)

## Additional information

### Funding

| Funder | Grant reference number | Author |
| --- | --- | --- |
| Starry Night Science Fund of Zhejiang University Shanghai Institute for Advanced Study | SN-ZJU-SIAS-005 | Liquan Huang |
| Starry Night Science Fund of Zhejiang University Shanghai Institute for Advanced Study | SN-ZJU-SIAS-003 | Ruhong Zhou |
| National Key Research and Development Program of China | 2021YFF1200803 | Liquan Huang |
| National Key Research and Development Program of China | 2021YFF1200404 | Ruhong Zhou |
| National Key Research and Development Program of China | 2021YFA1201200 | Ruhong Zhou |
| National Natural Science Foundation of China | U1967217 | Ruhong Zhou |
| National Center of Technology Innovation for Biopharmaceuticals | NCTIB2022HS02010 | Ruhong Zhou |
| National Independent Innovation Demonstration Zone Shanghai Zhangjiang Major Projects | ZJZX2020014 | Ruhong Zhou |

The funders had no role in study design, data collection and interpretation, or the decision to submit the work for publication.

### Author contributions

Yi-Hong Li, Conceptualization, Data curation, Formal analysis, Validation, Investigation, Visualization, Methodology, Writing – original draft; Yi-Sen Yang, Formal analysis, Investigation, Visualization, Methodology, Writing – original draft; Yan-Bo Xue, Formal analysis, Validation, Investigation, Visualization, Methodology; Hao Lei, Data curation, Investigation, Visualization, Methodology; Sai-Sai Zhang, Yushi Yao, Investigation, Methodology; Junbin Qian, Investigation, Visualization; Ruhong Zhou, Conceptualization, Supervision, Funding acquisition, Writing - review and editing; Liquan Huang, Conceptualization, Resources, Data curation, Formal analysis, Supervision, Funding acquisition, Validation, Visualization, Methodology, Writing – original draft, Project administration, Writing - review and editing

### Author ORCIDs

Liquan Huang http://orcid.org/0000-0003-3400-0685

## Ethics

This study was performed in strict accordance with the recommendations in the Guide for the Care and Use of Laboratory Animals of the National Institutes of Health. All of the animals were handled according to approved institutional animal care and use committee (IACUC) protocol ( ZJU20240147) of Zhejiang University.

Reviewer #1 (Public Review): https://doi.org/10.7554/eLife.92956.3.sa1
Reviewer #2 (Public Review): https://doi.org/10.7554/eLife.92956.3.sa2
Author response https://doi.org/10.7554/eLife.92956.3.sa3

# Additional files

## Supplementary files
• MDAR checklist

## Data availability

All data generated or analysed during this study are included in the manuscript and supporting files; source data files have been provided for Figures 1-8.

The following previously published dataset was used:

| Author(s) | Year | Dataset title | Dataset URL | Database and Identifier |
|-----------|------|---------------|-------------|-------------------------|
| Vaughan AE | 2022 | Injury-induced pulmonary tuft cells are heterogenous, arise independent of key Type 2 cytokines, and are dispensable for dysplastic repair | http://www.ncbi.nlm.nih.gov/geo/query/acc.cgi?acc=GSE197163 | NCBI Gene Expression Omnibus, GSE197163 |

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
