## [Editor Report · eLife assessment]

This, in principle, **useful** study suggests that the G-protein subunit Gng13 is required for limiting injury and inflammation following H1N1 influenza infection via anti-inflammatory effects from ectopic tuft cells. While support for Gng13 helping to limit influenza injury in the transgenic mouse models used here is **solid**, evidence for these effects being mediated by normal tuft cells remains **incomplete**, giving conflicting data from mice that lack tuft cells entirely.

---

## [Referee Report · Reviewer #1 (Public Review)]

Li et al. report here on the expression of a G-protein subunit Gng13 in ectopic tuft cells that develop after severe pulmonary injury in mice. By deleting this gene in ectopic tuft cells as they arise, the authors observed worsened lung injury and greater inflammation after influenza infection, as well as a decrease in the overall number of ectopic tuft cells. This was in stark contrast to deletion of Trpm5, a cation channel generally thought to be required for all functional gustatory signaling in tuft cells, where no phenotype is observed. Strengths here include a thorough assessment of lung injury via a number of different techniques. Weaknesses are notable: Confusingly, these findings are at odds with reports from other groups demonstrating no obvious phenotype upon influenza infection in mice lacking the transcription factor Pou2f3, which is essential for all tuft cell specification and development. The authors speculate that heterogeneity within nascent tuft cell populations, specifically the presence of pro- and anti-inflammatory tuft cells, may explain this difference, but they do not provide any data to support this idea.

Notes on revision: The authors provided responses to some of my critiques. I think the central discrepancy between the lack of a phenotype in Pou2f3 and Trpm5 KO mice compared to the stronger phenotype in the Chat-Cre / Gng13 KO mice remains unresolved and will require future work to provide a clear model. This may or may not ultimately involve tuft cell heterogeneity.

---

## [Referee Report · Reviewer #2 (Public Review)]

Summary:

The study by Li et al. aimed to demonstrate the role of the G𝛾13-mediated signal transduction pathway in tuft cell-driven inflammation resolution and repairing injured lung tissue. The authors showed the reduced number of tuft cells in the parenchyma of G𝛾13 null lungs following viral infection. Mice with a G𝛾13 null mutation showed increased lung damage and heightened macrophage infiltration when exposed to the H1N1 virus. Their further findings suggested that lung inflammation resolution, epithelial barrier and fibrosis were worsen in G𝛾13 null mutants.

Strengths:

The revised study carefully analyzed phenotypes in mice lacking G𝛾13 in response to viral infection, providing further support that G𝛾13+ tuft cells play a role in the resolution of inflammation and injury repair.

---

## [Author Response]

The following is the authors’ response to the original reviews.

We have made revisions accordingly. The following is a list of the changes we have made in this revised Version of Record:

(1) We have added three more panels to Figure 1-figure supplement 1, showing that lipopolysaccharide-induced severe lung injury also generate some ectopic tuft cells expressing both Dclk1 and Gα-gustducin, a G protein α subunit expressed in taste bud cells and many tuft cells.

(2) We have added a new supplemental figure, Figure 2-figure supplement 1, showing the reanalysis data of the single-cell RNAseq dataset (GSE197163) indicating the numbers of Trpm5-GFP+ ectopic tuft cells expressing Tas2r108, Tas2r105, Tas2r138, Tas2r137 and other Tas2rs, respectively. And the original “Figure 2-figure supplement 1” in the previous version has been changed to “Figure 2-figure supplement 2”.

(3) We have added another new supplemental figure, Figure 3-figure supplement 1, showing the H1N1 infection-damaged lung tissue volumes in the Gng13-cKO mice are significantly greater than those in WT or Trpm-/- mice, which is in agreement with the data of the injured lung surface areas from these three genotypes of mice (Figure 3 C and D). And the original “Figure 3-figure supplement 1” in the previous version has been changed to “Figure 3-figure supplement 2”.

(4) We have added to the new Figure 3-figure supplement 2 two new panels: I and J, showing the reanalysis data of the single-cell RNAseq dataset (GSE197163), indicating that about 57% of Trpm5-GFP+ ectopic tuft cells express Gγ13, some of which express Alox5, a key enzyme to the biosynthesis of pro-resolving mediators.

(5) We have added one reference on Sytox and another on Alox5.

(6) We have corrected two labeling errors to Figure 3 G and M, and some other typos in the article. Also, we have removed “Present address” attached to some authors since no present address was needed at all.

Attached below is our point-by-point reply to the comments and suggestions made by the reviewers. We hope that you and the reviewers will find all concerns satisfactorily addressed.

**Responses to public reviews:**

**Reviewer #1:**
Li et al. report here on the expression of a G-protein subunit Gng13 in ectopic tuft cells that develop after severe pulmonary injury in mice. By deleting this gene in ectopic tuft cells as they arise, the authors observed worsened lung injury and greater inflammation after influenza infection, as well as a decrease in the overall number of ectopic tuft cells. This was in stark contrast to the deletion of Trpm5, a cation channel generally thought to be required for all functional gustatory signaling in tuft cells, where no phenotype is observed. Strengths here include a thorough assessment of lung injury via a number of different techniques. Weaknesses are notable: confusingly, these findings are at odds with reports from other groups demonstrating no obvious phenotype upon influenza infection in mice lacking the transcription factor Pou2f3, which is essential for all tuft cell specification and development. The authors speculate that heterogeneity within nascent tuft cell populations, specifically the presence of pro- and anti-inflammatory tuft cells, may explain this difference, but they do not provide any data to support this idea.

We thank the reviewer for pointing out the strengths of this work. The phenotypes of the Gng13 conditional knockout mice upon severe pulmonary injury seem to be severer than those of Trpm5 knockout or Pou2f3 knockout mice, which we would attribute to functionally specific tuft cell subtypes. In the intestines, tuft cells are known to promote type II innate immune responses. Those ectopic pulmonary tuft cells emerge at 12 days post infection, and may not be involved in the initial immune responses to the infection, and instead, some of them may contribute to the inflammation resolution and functional recovery. Reanalysis of the previously published single tuft cell RNAseq dataset indeed showed that Gng13 is expressed in a subset of these ectopic pulmonary tuft cells, and anti-inflammatory genes such as Alox5 are also found in some of these tuft cells (please see the newly added Figure 3 supplement 2 I and J). Together, these data suggest that while some of these tuft cells may still play a pro-inflammatory role as in the intestines, some other Gγ13-expressing tuft cells contribute to the inflammation resolution, and disruption of the latter’s function results in the severer phenotypes.

**Reviewer #2:**
The study by Li et al. aimed to demonstrate the role of the Gγ13-mediated signal transduction pathway in tuft cell-driven inflammation resolution and repairing injured lung tissue. The authors showed a reduced number of tuft cells in the parenchyma of Gγ13 null lungs following viral infection. Mice with a Gγ13 null mutation showed increased lung damage and heightened macrophage infiltration when exposed to the H1N1 virus. Their further findings suggested that lung inflammation resolution, epithelial barrier, and fibrosis were worsened in Gγ13 null mutants.Strengths:The beautiful immunostaining findings do suggest that the number of tuft cells is decreased in Gr13 null mutants.Weaknesses:The description of phenotypes, and the approaches used to measure the phenotypes are problematic. Rigorous investigation of the mouse lung phenotypes is needed to draw meaningful conclusions.

Thank the reviewer for pointing out the major findings and strengths of our work. Regarding the approaches used to measure the phenotypes, we first did double immunostaining and validated that the lipopolysaccharide-induced DCLK1+ positive cells are indeed ectopic pulmonary tuft cells with an antibody to Gα-gustducin, a commonly expressed G protein α subunit in taste buds and tuft cells. Second, in addition to the measurements of the injured lung surface areas, we determined the injured lung tissue volumes by slicing the injured lungs into a series of tissue sections, quantifying the injured areas in each section and then reconstructing the injured volumes. Third, we reanalyzed the previously published single-tuft cell RNAseq dataset and found that a subset (i.e., ~57%) of Trpm5-GFP+ tuft cells express Gng13, some of which express anti-inflammatory genes such as Alox5. These additional data further support our finding that a subset of these Gγ13-expressing ectopic tuft cells may contribute to the inflammation resolution while others may play a proinflammatory role.

**Reply to the recommendations of Reviewer #1:**
(1) A major issue with this study is the fact that Chat-Cre mediated knockout of Gng13 leads to reduced tuft cells and impaired recovery, yet global TRPM5 deletion (this study) and global Pou2f3 deletion (Barr et al.) exhibit no apparent phenotype. One can imagine a Trpm5-independent role of Gng13 in tuft cells, but it is much harder to reconcile with the fact that Pou2f3 KO mice, which lack tuft cells entirely, exhibit no apparent phenotype. This was examined in some detail in Barr et al., demonstrating no apparent change in weight loss, dysplastic expansion (Krt5+ cells), or goblet cell metaplasia. The most parsimonious explanation is that Gng13 deletion in another Chat+ cell type, probably neurons of some sort, is leading to this phenotype. The authors really need to investigate this in some detail as the data does not really support a role of tuft cells in the phenotype they observe. Better yet, identification of the other Chat+ cell type in which Gng13 deletion promotes impaired lung recovery would be very interesting. While neurons seem likely, perhaps there is another Chat+ cell type expressing Gng13 in the respiratory tract that could be playing a role as well. In either case, the discrepancy between Pou2f3 KO (no phenotype) and Chat-Cre / Gng13 KO (impaired recovery) is difficult to reconcile.

We agree with the reviewer, and it took us some time to make senses of the data as well. The differences in phenotypes between Trpm5-knockout versus Gng13 conditional knockout (Gng13-cKO) could be explained by that Gγ13 is a partner of Gβγ moiety of a heterotrimeric G protein (Gαβγ)，which is known to act on many effector enzymes and ion channels, while Trpm5 largely regulates the influx of monovalent cations, depolarizing the plasma membrane potentials. Thus, it is understandable that nullification of Gng13 may have more profound effect on cell physiology and consequent phenotypes than that of Trpm5, and similar differential effects were also found in the intestines (Frontiers in Immunology, 2023, DOI 10.3389/fimmu.2023.1259521).

Data from several research groups have indicated that there are subtypes of tuft cells, each of which displays unique gene expression patterns as well as input and out signal profiles. It is yet not well understood how each subtype may contribute to the inflammatory responses or inflammation resolution. Comparative analyses of our data from the Gng13-cKO mice versus those from Pou2f3-KO mice suggest that Gng13-expressing tuft cells may have a role in the inflammation resolution while other ectopic tuft cells may contribute to the maintenance of the inflammation at a certain level, impairing subsequent tissue repairing and recovery. The exact molecular and cellular mechanisms are to be revealed in our future studies.

The central nervous system may also play a role in the impaired lung recovery. But our detailed immunochemical studies did not identify any significant number of neurons innervating the lung tissue co-expressing ChAT and Gng13, suggesting that no immediate action from these neurons may regulate the pulmonary inflammation resolution or functional recovery.

Together, our data suggest the importance of tuft cell subtype-specific functions, which may help us further understand the role of these rare tuft cells.

(2) Figures showing alternative injury models inducing the generation of ectopic tuft cells are not convincing and not quantified. DCLK1 can be a bit promiscuous, so verifying tuft cell expansion in these other models with other markers (especially for LPS and HDM which have not been reported elsewhere) is important.

We agree with the reviewer that DCLK1 is not a very specific marker for tuft cells. We have also observed that chemical inductions of these ectopic tuft cells with bleomycin, HDM or LPS are not as effective as H1N1 viruses. To verify that these rare DCLK1-positive cells are indeed tuft cells, we performed double immunostaining with antibodies to DCLK1 and to Gα-gustducin, another tuft cell marker. The results showed that some of these spindle-shaped DCLK1 positive cells indeed also express Gα-gustducin (see the newly added panels in Figure 1-figure supplement 1), indicating that they are most likely the chemically induced ectopic tuft cells. We also agree with the reviewer that it would be important to further investigate the possible roles of these cells during the stages of the chemically induced injury, inflammation resolution and functional recovery.

(3) Calcium responses in isolated post-flu tuft cells are interesting but difficult to interpret as presented. Can higher-power images be shown? Also, no statistical analysis is presented to provide any confidence in that data.

Thank the reviewer for the suggestions. As found in taste buds, only a subset of these ectopic tuft cells expresses Tas2rs, and each of these cells may express a few of the 35 murine Tas2rs. Thus, a particular bitter tasting compound can activate only few tuft cells and we had to use low-magnification to include more responsive cells in a field under the imaging microscope. We agree with the reviewer that it would be an interesting idea to statistically correlate the response profile to bitter substances with the cell’s Tas2r expression pattern, which we have done with sperm cells before (Molecular Human Reproduction, 2013, doi:10.1093/molehr/gas040). However, the main focus of this work is on the effect of Gng13-cKO in a subset of these ectopic tuft cells on the recovery. We plan to investigate these interesting cells in more details in the future.

(4) I am unaware of Sytox being a specific dye for pyroptotic cells. Can the authors please provide a reference or otherwise justify this?

Sytox is a dye to stain dead cells, which has been used previously in the studies on gasdermin-mediated lytic cell death (Xi et al., Up-regulation of gasdermin C in mouse small intestine is associated with lytic cell death in enterocytes in worm-induced type 2 immunity. PNAS 2021 118(30) e2026307118 https://doi.org/10.1073/pnas.2026307118). In our work we used the dye for the same assay.

(5) The authors perform qPCR for various taste receptor genes pre- and post-flu, but do not show that these genes are specifically induced in tuft cells. Since single-cell data and bulk RNA-Seq are available from Barr et al., the authors should validate the expression of these Tas2r genes specifically in post-flu tuft cells.

Thank the reviewer for the suggestion. Yes, we have performed analysis of the single-cell RNAseq dataset (GSE197163, Barr et al. 2022) and found that among 613 Trpm5-GFP+ tuft cells, Tas2r108 was expressed in the greatest number of cells, i.e., 67 cells, followed by Tas2r105, Tas2R138, Tas2r137, Tas2r118 and Tas2r102, which were detected in 11, 10, 10, 5 and 4 cells, respectively (see the newly added Figure 2-figure supplement 1). This order of expressing cell numbers is very much in agreement with that of the relative Tas2r expression levels obtained with the qPCR experiment (Figure 2A), indicating the expression of these Tas2rs likely in the ectopic tuft cells. We will further validate the data by analyzing the bulk RNA-Seq dataset when it is accessible to us.

(6) Some general editing of language throughout would be helpful to increase readability.

Thanks for pointing out. We have carefully checked the manuscripts, corrected some typos and revised several sentences to increase its readability.

(7) For the fibrosis analysis, trichrome staining is very heterogenous, which is reflected by the large error bars in Fig. 8B. A more quantitative, "whole lung" analysis such as hydroxyproline content or western blotting for Col1a1 would be ideal.

The approach of Masson’s trichrome staining along with qRT-PCR assays on the fibrotic gene expression has been used previously to quantitatively analyze fibrosis (e.g., Zhang et al., Neuropilin-1 mediates lung tissue-specific control of ILC2 function in type 2 immunity. Nature Immunology 23:237-250, 2022, https://doi.org/10.1038/s41590-021-01097-8). We agree with the reviewer that there are large error bars in Fig. 8B, and hydroxyproline content assay or western blotting for Col1a1 would be ideal. But our qRT-PCR was performed on the RNA samples extracted from the “whole lungs”, and its data are also able to reflect the extent of fibrosis of the lungs.

(8) The authors claim that only a subset of tuft cells express Gng13, but this is supported only by a single IF image in Fig. 3 supplement 1G. The authors could download the single-cell dataset from Barr et al. to confirm the heterogeneity of Gng13 expression and get a better sense of the fraction of total ectopic tuft cells that express this, as it is a critical point in their model.

Thank the reviewer for the suggestion. Yes, we have downloaded and reanalyzed the single-cell RNAseq dataset (GSE197163), and found that out of 613 Trpm5-GFP+ tuft cells, 350 or 57% of these cells expressed Gng13 (Figure 3-figure supplement 2I). This result, together with our immunohistochemical data (Figure 3-figure supplement 2G and H) indicates that Gγ13 is expressed in a subset of these ectopic tuft cells. More comprehensive studies are needed to characterize these tuft cell subtypes and elucidate subtype-selective functions.

**Reply to the recommendations of Reviewer #2:**
The study needs more rigorous examinations of the phenotypes. For example, quantification of the injury area in Fig3C is problematic. Similarly, fibrotic phenotype and quantification in Fig 8C also have problems. This study heavily used qRT-PCR analysis to quantitate the level change of bitter/other receptors in a minor population of tuft cells which are also minor in a whole lung. Given the limited number of cells, it is difficult to appreciate that qRT-PCR can pick up the difference. In addition, how would the findings in this study reconcile with the finding by Huang (PMID: 36129169) where pou2f3 null mutants (without tuft cells) were used? Huang et al. did not observe more severe phenotypes in the mice without tuft cells than controls.

Thank the reviewer for the recommendations. Regarding Fig 3C, please see the reply below: revisions for clarity point #2.

Fig 8 B and C used Masson’s trichrome staining to quantitatively analyze fibrosis, which has been used by other groups as well (e.g., Zhang et al., Neuropilin-1 mediates lung tissue-specific control of ILC2 function in type 2 immunity. Nature Immunology 23:237-250, 2022, https://doi.org/10.1038/s41590-021-01097-8). Our qRT-PCR data on the fibrotic gene expression (Figure 8A) further support the Masson’s trichrome staining results.

We realized that tuft cells make up only a minor population in the lungs. So, we performed qRT-PCR assays on the RNA samples isolated from mostly the injured tissues along with the corresponding tissues from the uninjured lungs as control. To validate our qRT-PCR data, we reanalyzed the previously published single ectopic tuft cell RNAseq dataset (GSE197163), and found that the most abundantly expressed Tas2r108 determined by qRT-PCR was also expressed in the greatest number of tuft cells, and the order of expression levels of other Tas2rs are also well in agreement between the qRT-PCR and single-cell RNAseq data (Figure 2A, Figure 2-figure supplement 1), cross-validating the data obtained by these two very different approaches.

We have carefully studied the finding by Huang (PMID: 36129169). Our data suggest that there are subtypes of the ectopic tuft cells, some of which contribute to the inflammation resolution while others play a proinflammatory role. Indeed, the reanalysis of the aforementioned single tuft cell RNAseq dataset found that about 57% Trpm5-GFP+ ectopic tuft cells expressed Gng13, and some of which expressed Alox5, a key enzyme to the biosynthesis of pro-resolving mediators. Thus, in the Pou2f3-knockout mice, both pro- and anti-inflammatory tuft cells are ablated, it would be hard to observe any significant phenotypes. When the function of a subset of Gγ13-expressing tuft cells is disrupted, the anti-inflammatory role from these cells is eliminated, resulting severer phenotypes. More studies are needed to further understand the subtype-specific functions of these fascinating tuft cells.

Do Gγ13 null mutants show similar phenotypes in bleomycin injury model?

Bleomycin and other chemicals-induced injury models indeed engender much fewer ectopic pulmonary tuft cells. Thus, it is more difficult to test the effect of Gng13 mutation due to the small number of the Gng13-expressing tuft cells in either WT or mutant lungs.

What is the cell fate of lineage labeled tuft cells in the lungs of Chat-Cre:Ai9:Gng13flox/flox mice following viral infection at different times examined? The numbers were decreased at different time points post-injury based on the data. Did these cells undergo apoptosis?It is an excellent idea to look into the cell fate of ChAT-Cre:Ai9:Gng13flox/flox. We believe that these cells would have a similar fate to other ectopic tuft cells, probably undergoing apoptosis. But our data suggest that Gng13 mutation suppresses the increase the ectopic tuft cells, or the increase of a particular subtype of these tuft cells. Further studies are needed to elucidate the molecular mechanisms of the Gγ13-mediated signal transduction pathways regulating the proliferation of a subset of ectopic tuft cells.Here are the revisions for clarity and coherence to the figures:(1) Fig 2: For the functional assessment, using tracheal tuft cells from the same ChAT-Cre:Ai9 mice would be a suitable positive control in the calcium response traces experiment. These specific cells could also serve as a control in Fig2a.

We would agree with the reviewer that tracheal tuft cells from the same ChAT-Cre: Ai9 mice would be an ideal positive control in the calcium response experiment as well as in the qRT-PCR assay. But we have established reliable methods to calcium image primary cells expressing taste receptors and quantify their RNA expression levels, which have been used in our previous publications, e.g., (1) Functional characterization of bitter taste receptors expressed in mammalian testis. Molecular Human Reproduction, 2013, doi:10.1093/molehr/gas040; (2) Infection by the parasitic helminth Trichinella spiralis activates a Tas2r-mediated signaling pathway in intestinal tuft cells. PNAS 2019, https://www.pnas.org/cgi/doi/10.1073/pnas.1812901116. We thank the reviewer for the excellent suggestion.

(2) Fig 3C: It is not clear whether the depicted areas really represent the injured area. To provide a more comprehensive view, the authors should also provide histological analysis and quantification of the injured lung. A 3D representation of the injury area would offer a more accurate presentation.

Thank the reviewer for the point. The depicted areas in Fig 3C are indeed the injured surface areas of the lungs. Following the reviewer’s suggestion, we carried out the histological analysis to determine the injured tissue volumes of the lungs. We fixed the lungs, and sliced them into 12 μm-thick sections, which were imaged under a microscope. The injured areas in a section were identified and quantified using the ImageJ software, and then the injured volume for this section was obtained by multiplying the area by the thickness of the section, i.e., 12 μm. Statistical analyses indicate that the injured volume of the Gng13-cKO lungs is significantly more than those of WT or Trpm5-KO mice, which has been included in Figure 3-figure supplement 1, and is in agreement with the data of the injured surface areas (Fig 3C).

(3) Fig 3 G/I/K/M: There seems to be an inconsistency in the time points. There's no indication for 14 dpi, yet two for 25 dpi. Additionally, a color legend for each sample would be helpful.

Thank the reviewer for pointing out. There were two typos, which have been corrected. Yes, the time points should be 14 dpi, 20 dpi, 25 dpi and 50 dpi. And a color legend has been added as well.

(4) Fig 4A: Using CD64 co-stained with Krt5 might better highlight the immune cells in the damaged region. Additionally, could you clarify the choice of the neutrophil marker CD64 over CD45 for staining the injured lung?

We agree with the reviewer that Krt5 antibody staining can help define the damaged region. We sectioned the lung tissues with a special attention to the damaged areas, but we found that the adjacent healthy areas also had extra immune cells. Thus, we counted in all these CD64+ cells in both the damaged as well as the surrounding, seemingly healthy areas. We used CD64 instead of CD45 to label these altered immune cells because we found that CD64 can better label the differential immune cells between WT and Gng13-cKO mice following H1N1 infection. Furthermore, CD64-labeled cells could be readily related to the Gsdmd/Gsdme-expressing F4/80-labeled immune cells shown in Figure 5 and its supplemental figures.

(5) Fig 5 and Supplemental Fig 5: It appears that the F4/80 staining exhibits notable background staining.

Yes, there is some background staining. The antibody was the best we could find, but its quality could be further improved. On the other hand, we thought that there were some cellular debris that might be stained positive by that antibody. At a higher magnification, however, we could still identify individual cells co-expressing IL-1β.

(6) Fig 8C: The depicted area does not seem to adequately represent the fibrosis in the injured lung.

Masson’s trichrome staining has been previously used to quantitatively analyze fibrosis (e.g., Zhang et al., Neuropilin-1 mediates lung tissue-specific control of ILC2 function in type 2 immunity. Nature Immunology 23:237-250, 2022, https://doi.org/10.1038/s41590-021-01097-8). Our qRT-PCR assays on the fibrotic gene expression (Figure 8A) were performed on the RNA samples extracted from the whole lungs, and the resultant data are able to reflect the extent of fibrosis of the lungs, although we also agree with the reviewer that additional data would make the conclusion more convincing.